# *Fn*-OMV potentiates ZBP1-mediated PANoptosis triggered by oncolytic HSV-1 to fuel antitumor immunity

Shuo Wang [1,3], An Song[1,3], Jun Xie[1,2], Yuan-Yuan Wang[1], Wen-Da Wang[1], Meng-Jie Zhang[1], Zhi-Zhong Wu[1], Qi-Chao Yang[1], Hao Li [1], Junjie Zhang [1,2] ✉ & Zhi-Jun Sun [1] ✉

Oncolytic viruses (OVs) show promise as a cancer treatment by selectively replicating in tumor cells and promoting antitumor immunity. However, the current immunogenicity induced by OVs for tumor treatment is relatively weak, necessitating a thorough investigation of the mechanisms underlying its induction of antitumor immunity. Here, we show that HSV-1-based OVs (oHSVs) trigger ZBP1-mediated PANoptosis (a unique innate immune inflammatory cell death modality), resulting in augmented antitumor immune effects. Mechanistically, oHSV enhances the expression of interferon-stimulated genes, leading to the accumulation of endogenous Z-RNA and subsequent activation of ZBP1. To further enhance the antitumor potential of oHSV, we conduct a screening and identify *Fusobacterium nucleatum* outer membrane vesicle (*Fn*-OMV) that can increase the expression of PANoptosis execution proteins. The combination of *Fn*-OMV and oHSV demonstrates potent antitumor immunogenicity. Taken together, our study provides a deeper understanding of oHSV-induced antitumor immunity, and demonstrates a promising strategy that combines oHSV with *Fn*-OMV.

Oncolytic viruses (OVs) have garnered significant attention in the field of cancer therapy due to their unique ability to specifically target and eliminate tumors[1–5]. Recently, the emphasis in OVs therapy has shifted to focus on their immunostimulatory effects[6,7]. Research suggests that OVs can induce antitumor immune responses by releasing tumor-associated antigens and inflammatory signals[8,9]. Furthermore, OVs have shown promise in improving the efficacy of immune checkpoint blockade (ICB) therapy[10]. Clinical trials exploring the use of OVs alone or in combination with other therapies, including ICB, have yielded encouraging results[10–12]. However, OVs still face challenges in terms of their efficacy, among other issues. Enhancing the antitumor immunogenicity induced by OVs is crucial for improving their therapeutic

effectiveness[13,14]. Therefore, further elucidating the mechanisms underlying OVs-induced antitumor immunity and exploring strategies to enhance their immunogenicity are important scientific challenges that need to be addressed.

Importantly, OVs, as pathogenic microorganisms, have a close association with innate immune sensors such as inflammasomes, which play a critical role in eliciting antitumor immune responses[15,16]. The mechanisms underlying the induction of innate immune responses by herpes simplex virus-1 (HSV-1) have been extensively studied[17,18], leading to significant advances in the development of HSV-1-based OVs (oHSVs) for cancer treatment[1,11]. Previous research has shown that HSV-1 infection induces Z-DNA binding protein 1 (ZBP1)-mediated cell death

[1]State Key Laboratory of Oral & Maxillofacial Reconstruction and Regeneration, Key Laboratory of Oral Biomedicine Ministry of Education, Hubei Key Laboratory of Stomatology, School & Hospital of Stomatology, Frontier Science Center for Immunology and Metabolism, Taikang Center for Life and Medical Sciences, Wuhan University, Wuhan 430079, China. [2]Hubei Key Laboratory of Tumor Biological Behaviors, Hubei Province Cancer Clinical Study Center, Zhongnan Hospital of Wuhan University, Wuhan 430071, China. [3]These authors contributed equally: Shuo Wang, An Song. ✉e-mail: junjiezhang@whu.edu.cn; sunzj@whu.edu.cn

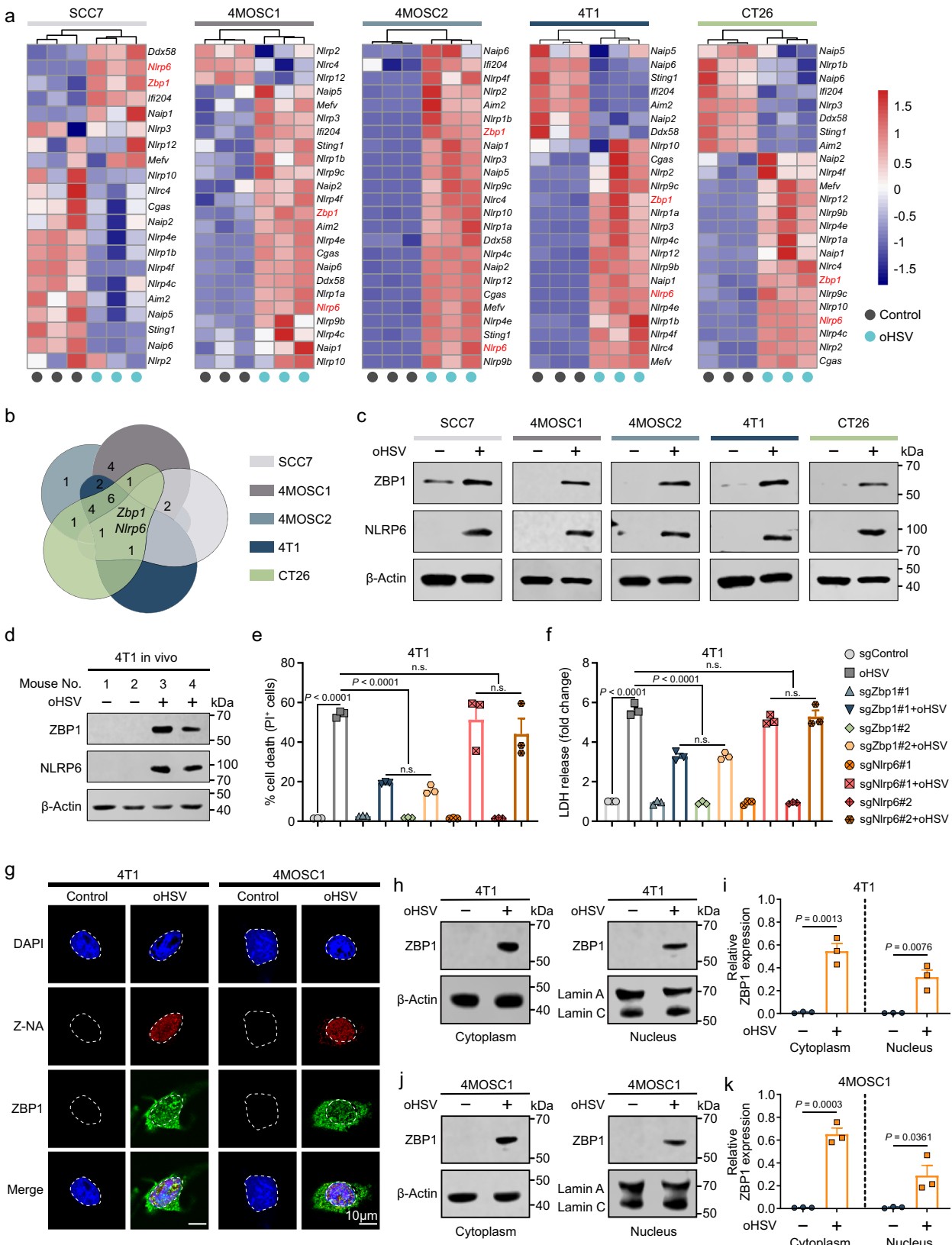

and the release of inflammatory cytokines, playing a vital role in the regulation of innate immunity[19]. ZBP1, acting as a sensor for Z-type nucleic acids (Z-NA), can detect both endogenous and exogenous Z-NA through its Zα domain and subsequently trigger downstream cell death[20]. Z-NA is a distinct left-handed double-helical structure of nucleic acids that plays a critical role in various diseases, including

cancer[21,22]. However, it is currently unclear whether and how Z-NA is generated during HSV-1 infection. In addition, there have been no reports on whether oHSV can exert antitumor immune effects by activating ZBP1 in tumors.

PANoptosis is defined as a unique inflammatory programmed cell death pathway that is driven by caspases and receptor-interacting

**Fig. 1 | oHSV induces ZBP1-mediated tumor cell death.** Heatmap showing the expression of signature genes related to inflammasomes and certain key innate sensor-related genes in five types of tumor cells infected with oHSV (MOI = 1, 24 h) (**a**) and Venn diagram showed the increased gene intersections of five types of tumor cells infected with oHSV. Specifically, the two genes in the center indicate that two genes, *Zbp1* and *Nlrp6*, are commonly upregulated in all five cell lines infected with oHSV (**b**). **c** Five types of tumor cells were infected with oHSV (MOI = 1, 24 h), and cellular protein was analyzed by western blotting to detect the expression level of ZBP1 and NLRP6. Western blotting was done thrice independently with similar results. **d** oHSV-treated or untreated 4T1 tumor tissue lysates were analyzed of the expression of ZBP1 and NLRP6 by western blotting. Western blotting was done thrice independently with similar results. **e** Effect of ZBP1 and NLRP6 expression on relative cell viability in the 4T1 cells treated with oHSV

(MOI = 1, 24 h). n.s. not significant. **f** The effect of ZBP1 and NLRP6 expression on the injury of 4T1 cells after oHSV (MOI = 1, 24 h) treatment was detected by LDH release assay. n.s. not significant. **g** The expression of Z-NA and ZBP1 in tumor cells after oHSV (MOI = 1, 24 h) treatment was evaluated using a Confocal Laser Scanning Microscope. Scale bars = 10 μm. **h, i** Western blotting analysis assessing ZBP1 expression in the cytoplasm and nucleus of 4T1 cells with or without oHSV treatment (MOI = 1, 24 h). Western blotting was done thrice independently with similar results. **j, k** Western blotting analysis assessing ZBP1 expression in the cytoplasm and nucleus of 4MOSC1 cells with or without oHSV treatment (MOI = 1, 24 h). Western blotting was done thrice independently with similar results. *n* = 3 independent experiments for **a, e, f**. Statistical significance was determined using two-tailed Student's *t* test in **e, f, i, k**. Data represent the mean ± s.e.m. Scale bar, 10 μm. Source data are provided in the Source Data file.

protein kinases (RIPKs) and regulated by multiprotein PANoptosome complexes[23,24]. PANoptosis plays a significant role in tumor treatment[25,26], as this form of inflammatory cell death leads to the release of inflammatory signaling molecules and intracellular tumor antigens, triggering a robust immune response[27,28]. Research has revealed that ZBP1 is a key upstream regulator of PANoptosis[19,29,30]. This suggests that oHSVs, if they activate ZBP1, may exert their antitumor effects on tumors through ZBP1-mediated PANoptosis. In addition, studies suggest that inducing PANoptosis in tumors may significantly enhance antitumor immunity[26]. Current research focusing on the regulation and activation of PANoptosis primarily revolves around innate immune sensors[19,31], indicating the great potential of using pathogenic microorganisms to induce PANoptosis for therapeutic purposes. The intratumoral bacterial injection has gained significant attention as the intratumoral microbiota is closely associated with tumor prognosis[32,33]. However, due to the replicative capacity of bacteria, intratumoral injection of live bacteria poses a considerable risk of uncontrolled infection and inflammatory reactions. In contrast, bacterial outer membrane vesicles (OMVs) derived from bacteria retain the highly immunogenic properties of bacteria and possess good biosafety[34,35]. Previous studies have shown promising therapeutic effects when used OMVs alone or in combination with other drugs for cancer treatment. However, whether there is a synergistic therapeutic effect and the underlying mechanisms between bacterial OMVs and OVs remain unclear.

Here, we show that oHSV may exert antitumor effects through the induction of ZBP1-mediated PANoptosis. Moreover, the study indicates that oHSV promotes the upregulation of ZBP1 expression in tumor cells by generating endogenous Z-RNA. The investigation also reveals that *Fusobacterium nucleatum*-OMV (*Fn*-OMV) can upregulate the expression of the PANoptosis executor proteins gasdermin D/E (GSDMD/E) and mixed lineage kinase domain-like protein (MLKL) by interfering with ubiquitination processes. In addition, this study reveals that the combination strategy of *Fn*-OMV with oHSV harbors potent antitumor immune effects. Overall, through the key point of PANoptosis executor proteins, this study links oHSV with *Fn*-OMV. On the one hand, oHSV promotes the activation of PANoptosis executor proteins, while on the other hand, *Fn*-OMV upregulates the expression levels of PANoptosis executor proteins. The two possess distinct modes of action, yet they complement each other and generate positive feedback effects that amplify the therapeutic outcomes of OVs.

## Results

### Upregulation of ZBP1 by oHSV induces tumor cell death and triggers the release of immunogenic substances

Innate immunity, the line of defense against pathogens and host tissue damage, relies on inflammasomes for its crucial role[36]. Heatmap showing the mRNA expression levels of signature genes related to inflammasomes and certain key innate sensor-related genes in five types of tumor cells infected with oHSV (MOI = 1, 24 h) (Fig. 1a).

The Venn diagram depicts which genes were upregulated across multiple cell types versus upregulated in only certain cell types upon the addition of oHSV (Fig. 1b). Specifically, the two genes in the center indicate that two genes, *Zbp1* and *Nlrp6*, are commonly upregulated in all five cell lines infected with oHSV (Fig. 1b). Furthermore, the number "1" on the far left side signifies that one gene is upregulated only in the 4MOSC2 cell line but not in the other four cell types (Fig. 1b). The results demonstrated that only the transcription levels of *Zbp1* and *Nlrp6* were upregulated in all five cell lines upon oHSV infection (Fig. 1a, b). Western blotting analysis further confirmed the upregulation of ZBP1 and NLRP6 at the protein level in all five cell lines upon oHSV infection (Fig. 1c). Consistent with the findings in tumor cell lines, we also observed the upregulation of ZBP1 and NLRP6 in 4T1 tumors treated with oHSV (Fig. 1d).

To further explore the potential role of elevated ZBP1 and NLRP6 expression in oHSV infection in tumor cells, we generated ZBP1-deficient 4T1 and 4MOSC1 cells, as well as NLRP6-deficient 4T1 and 4MOSC1 cells, using CRISPR/Cas9-based gene editing. These cells were then treated with oHSV. We observed that depletion of ZBP1 resulted in a significant reduction in the number of tumor cell deaths induced by oHSV compared to that in sgControl cells (Fig. 1e and Supplementary Fig. 1a). However, depletion of NLRP6 did not have a significant impact on cell death (Fig. 1e and Supplementary Fig. 1a). In addition, we observed that, compared to sgControl cells, depletion of ZBP1 led to a significant decrease in oHSV-induced LDH release (Fig. 1f and Supplementary Fig. 1b) and ATP release (Supplementary Fig. 1c, d), whereas depletion of NLRP6 had no significant effect on ATP release or LDH release.

To further investigate the generation of Z-NA during oHSV infection in tumor cells, we employed cell immunofluorescence assays using Z-DNA/RNA antibodies. Our findings revealed a significant accumulation of Z-NA in SCC7, 4T1, 4MOSC1, 4MOSC2, and CT26 cells after 24 h of oHSV infection, accompanied by the upregulation of ZBP1 (Fig. 1g and Supplementary Fig. 2a). In addition, the augmented expression of ZBP1 in the cell nucleus was further validated through nuclear protein isolation (Fig. 1h–k).

### oHSV upregulates ZBP1 by inducing the accumulation of Z-RNA

Z-NA has two primary origins: exogenous, predominantly from some viral infections[37], and endogenous, primarily derived from dsRNA generated by endogenous retroviral elements (EREs)[38]. Under normal conditions, adenosine deaminase RNA specific 1 (ADAR1) inhibits the unintended accrual of endogenous Z-RNA through performing adenosine-to-inosine (A to I) edits, preventing recognition by nucleic acid sensors and thereby promoting the immune response[21,39]. However, in certain scenarios where ADAR1 editing function is impaired or when there is excessive accumulation of Z-NA override ADAR1 inhibitory capacity, the Z-NA accrual can activate ZBP1[21,40].

Notably, excessively generated ISG mRNAs, particularly those with a high editing index, can form Z-RNAs within the 3' UTRs, thus activating ZBP1[21]. We examined the expression of ISG mRNAs in 4T1

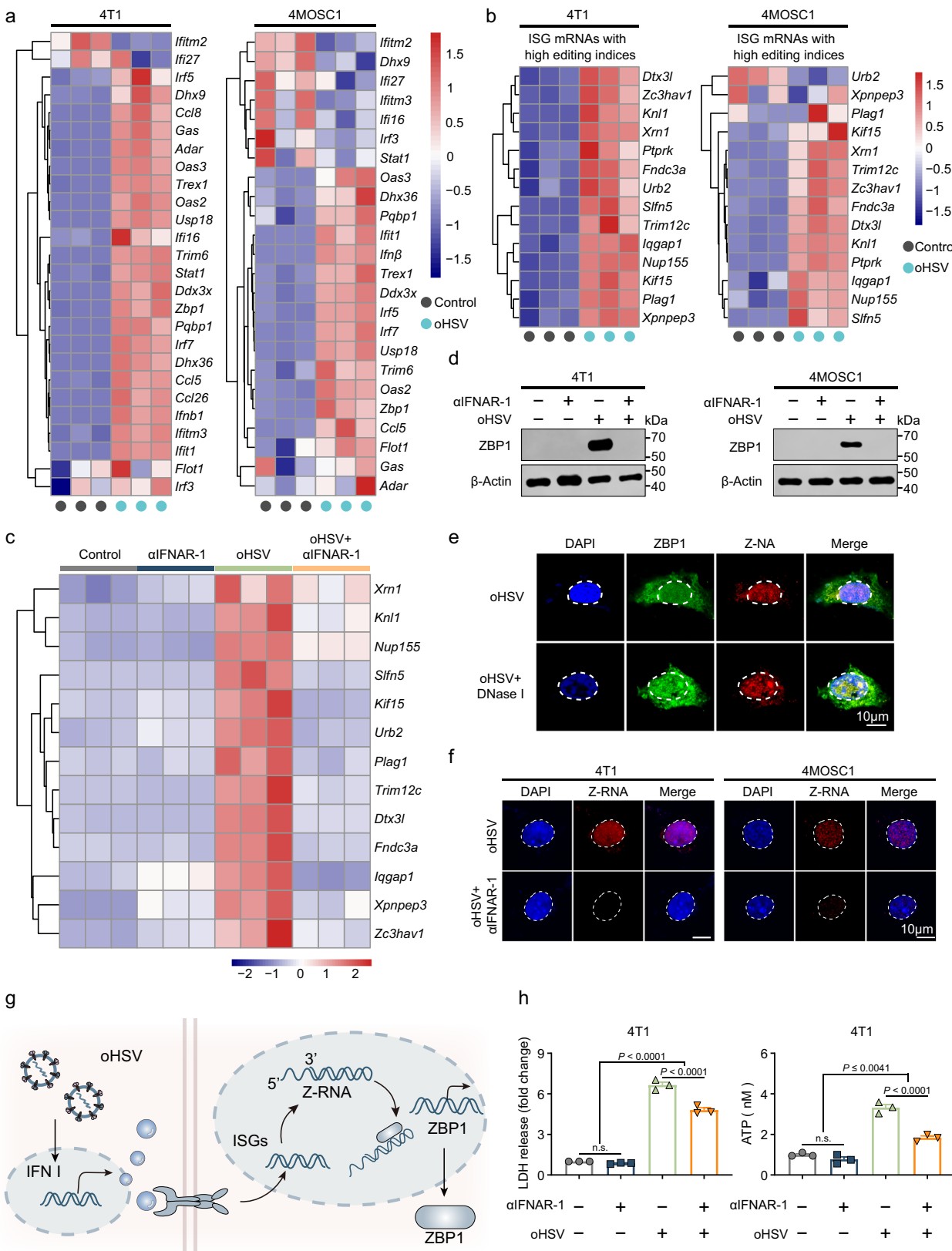

and 4MOSC1 cells during oHSV infection and observed a significant upregulation of ISG mRNAs, along with an elevation of ISG mRNAs containing Z-prone regions within the 3'-UTR (Fig. 2a, b). Furthermore, when IFNAR-1 was blocked using neutralizing antibodies, the expression of ISG mRNAs no longer increased (Fig. 2c). Western blotting analysis confirmed that the addition of neutralizing antibodies

completely inhibited the expression of ZBP1 (Fig. 2d). We initiated our protocol by incorporating DNase I to decompose DNA, followed by fluorescence detection, which revealed that DNase I treatment did not substantially alter the fluorescence signals of the Z-NA. This indicates that the accumulation of Z-NA is largely attributed to Z-RNA rather than Z-DNA (Fig. 2e). These results strongly suggest that during oHSV

**Fig. 2 | oHSV upregulates ZBP1 by inducing the accumulation of Z-RNA.**
**a** Heatmap showing the expression of signature genes related to ISG mRNA in oHSV-treated (MOI = 1, 24 h) 4T1 (left) and 4MOSC1 (right) cells. **b** Heatmap indicated the expression of signature genes related to ISG mRNA with high editing indices in oHSV-treated (MOI = 1, 24 h) 4T1 (left) and 4MOSC1 (right) cells. **c** Heatmap demonstrated the influence of the expression of ISG mRNA with high editing indices on oHSV-treated (MOI = 1, 24 h) 4T1 cells after IFNAR-1 blockade. **d** 4T1 cells were infected with oHSV (MOI = 1, 24 h) and IFNAR-1 blockade, the cellular protein was analyzed by western blotting to detect the expression level of ZBP1. Western blotting was done thrice independently with similar results. **e** The expression of Z-NA and ZBP1 in 4T1 tumor cells after oHSV (MOI = 1, 24 h) alone and oHSV combined with DNase I (25 UmL⁻¹) treatment was evaluated using a Confocal Laser Scanning Microscope. Scale bars = 10 μm. Data were repeated thrice independently with similar results. **f** The expression of Z-RNA in tumor cells after oHSV (MOI = 1, 24 h) treatment and IFNAR-1 blockade was evaluated using a Confocal Laser Scanning Microscope. Scale bars = 10 μm. Data were repeated thrice independently with similar results. **g** Model diagram illustrating the upregulation of ZBP1 by oHSV through the induction of Z-RNA accumulation. The release of LDH (left) and ATP (right) from 4T1 (**h**) tumor cells after oHSV (MOI = 1, 24 h) treatment and IFNAR-1 blockade. *n* = 3 independent experiments for (**a–c**, **h**). Statistical significance was determined using two-tailed Student's *t* test in (**h**). Data represent the mean ± s.e.m. Scale bar, 10 μm. Source data are provided in the Source Data file.

infection, the upregulation of ISG mRNAs, rather than the replication of viral nucleic acids, promotes the accumulation of endogenous Z-RNA, which upregulates ZBP1 (Fig. 2f, g). We observed that blockade of IFNAR-1 resulted in a reduced number of oHSV-induced tumor cell death (Supplementary Fig. 2b). Moreover, blockade of IFNAR-1 led to decreased oHSV-induced release of LDH and ATP (Fig. 2h and Supplementary Fig. 2c).

### oHSV exerts antitumor effects by triggering ZBP1-mediated PANoptosis in tumor cells

Our previous research revealed that oHSV induces cell death through the upregulation of ZBP1, which is considered a key regulator of PANoptosis and an innate immune sensor. To further elucidate the mechanism of oHSV-induced cell death, we generated GSDMD-, GSDME-, and MLKL-deficient 4T1 and 4MOSC1 cells using CRISPR/Cas9 gene editing (Supplementary Fig. 3a) and treated them with oHSV (MOI = 1, 24 h). Western blotting analysis demonstrated that oHSV induced ZBP1-mediated PANoptosis in tumor cells (Fig. 3a–d), along with the release of immunogenic substances (Fig. 3e and Supplementary Fig. 3b). In addition, we investigated the potential mechanisms by which oHSVs prevent tumor development in tumor-bearing mice. We observed that depletion of ZBP1 significantly inhibited the antitumor effects of oHSV, and depletion of GSDMD, GSDME, and MLKL also partially suppressed the antitumor effects of oHSV (Fig. 3f, g). These findings further suggest that oHSVs may exert antitumor effects on tumors through the induction of ZBP1-mediated PANoptosis.

### Fn-OMV enhances the expression levels of PANoptosis effector proteins by modulating protein ubiquitination

Due to the different biological characteristics of bacteria and viruses, the combined use of bacterial and viral therapies is believed to have a synergistic effect on tumor treatment[41]. Studies have reported that bacteria can release specific compounds to improve viral infection and replication, enhancing their ability to kill tumor cells[42]. Based on the microbial content and abundance in oral squamous cell carcinoma (OSCC) tumors and adjacent tissue samples (mucosa) tested in our previous study[43], we selected several bacteria that were abundant in tumors and unexpectedly discovered that *Fn* could increase the expression level of PANoptosis execution proteins in 4T1 cells (Fig. 4a, b). We observed similar effects in 4MOSC1 cells (Fig. 4b and Supplementary Fig. 4a). The effect was still observed even with prolonged exposure to the bacteria (Supplementary Fig. 4b).

Although it has been reported that intratumoral injection of *Fn* can enhance the therapeutic effect of ICB by increasing CD8⁺ tumor-infiltrating lymphocytes (TILs) in tumors, considering that *Fn* is usually considered a carcinogenic bacterium (which also suggests its proinflammatory ability), we further considered using *Fn*-OMV to observe whether similar effects could be obtained. *Fn*-OMV was extracted and characterized by transmission electron microscopy (Fig. 4c). We observed that 4T1 and 4MOSC1 tumor cells had a good phagocytic response to *Fn*-OMVs (Fig. 4d). We also tested the zeta potential and particle size of *Fn*-OMV (Fig. 4e and Supplementary Fig. 4c). We detected and observed that *Fn*-OMV could also increase the expression

level of PANoptosis execution proteins (Fig. 4f and Supplementary Fig. 4d), suggesting that *Fn* may enhance the expression level of PANoptosis execution proteins through *Fn*-OMV. We independently examined the effects of MG132 and *Fn*-OMV on the degradation of the GSDMD, GSDME, and MLKL proteins. Our results indicate that the degradation of GSDMD, GSDME, and MLKL occurs in the absence of MG132 stimulation, while MG132 significantly inhibits their degradation (Fig. 4g–i), suggesting that these proteins are degraded via the proteasome pathway. *Fn*-OMV also significantly inhibited the degradation of the GSDMD, GSDME, and MLKL proteins (Fig. 4g–i). Further experiments revealed that *Fn*-OMV suppresses the ubiquitination of these proteins (Fig. 4j–l). Therefore, we propose that *Fn*-OMV enhances the expression levels of these proteins by affecting their ubiquitination.

### Fn-OMV and oHSV have great potential for synergistic application

Previous studies reported that increasing the protein expression level of GSDME can transform cell apoptosis into pyroptosis, thereby enhancing the release of inflammatory substances[44]. Based on the discovery that *Fn*-OMV can increase the expression level of PANoptosis execution proteins, we hypothesized that *Fn*-OMV can enhance the ability of oHSV to induce PANoptosis in tumor cells, and further explored the potential synergistic mechanisms between *Fn*-OMV and oHSV. Flow cytometry results showed that compared to oHSV treatment alone, there was no significant difference in the number of dead tumor cells (Fig. 5a, b) and the release of LDH in tumor cells (Fig. 5c) after combined treatment with *Fn*-OMV and oHSV. However, the combined treatment of *Fn*-OMV and oHSV significantly increased the release of ATP in tumor cells compared to oHSV treatment alone (Fig. 5d). In addition, the combined treatment of *Fn*-OMV and oHSV significantly increased the release of nuclear high mobility group box 1 (HMGB1) (Fig. 5e) and tumor necrosis factor-alpha (TNF-α) (Fig. 5f) in tumor cells compared to oHSV treatment alone. We also tested the impact of combined oHSV and *Fn*-OMV treatment on the expression of PANoptosis executioners (Supplementary Fig. 5a). The results showed that compared to oHSV alone, combination of *Fn*-OMV with oHSV led to higher expression levels of PANoptosis executioners. These findings suggest that *Fn*-OMV may enhance the ability of oHSV to induce PANoptosis in tumor cells by increasing the expression level of PANoptosis execution proteins. Furthermore, we induced RAW264.7 cells to differentiate into M2 macrophages and separately added oHSV and *Fn*-OMV. The results showed that *Fn*-OMV had a significantly greater ability than oHSV to transform M2 macrophages into M1 macrophages (Fig. 5g, h). More importantly, we observed that *Fn*-OMV and oHSV had significantly different effects on the activation of inflammation-related genes in M2 macrophages (Fig. 5i, j). This suggests that there is great potential for the combined application of *Fn*-OMV and oHSV in tumors.

### The combination of Fn-OMV and oHSV induces an inflammatory gene signature and converts "cold" tumors into "hot" tumors

To further assess the effectiveness of combining *Fn*-OMV and oHSV for antitumor treatment, we carried out in vivo experiments. The results

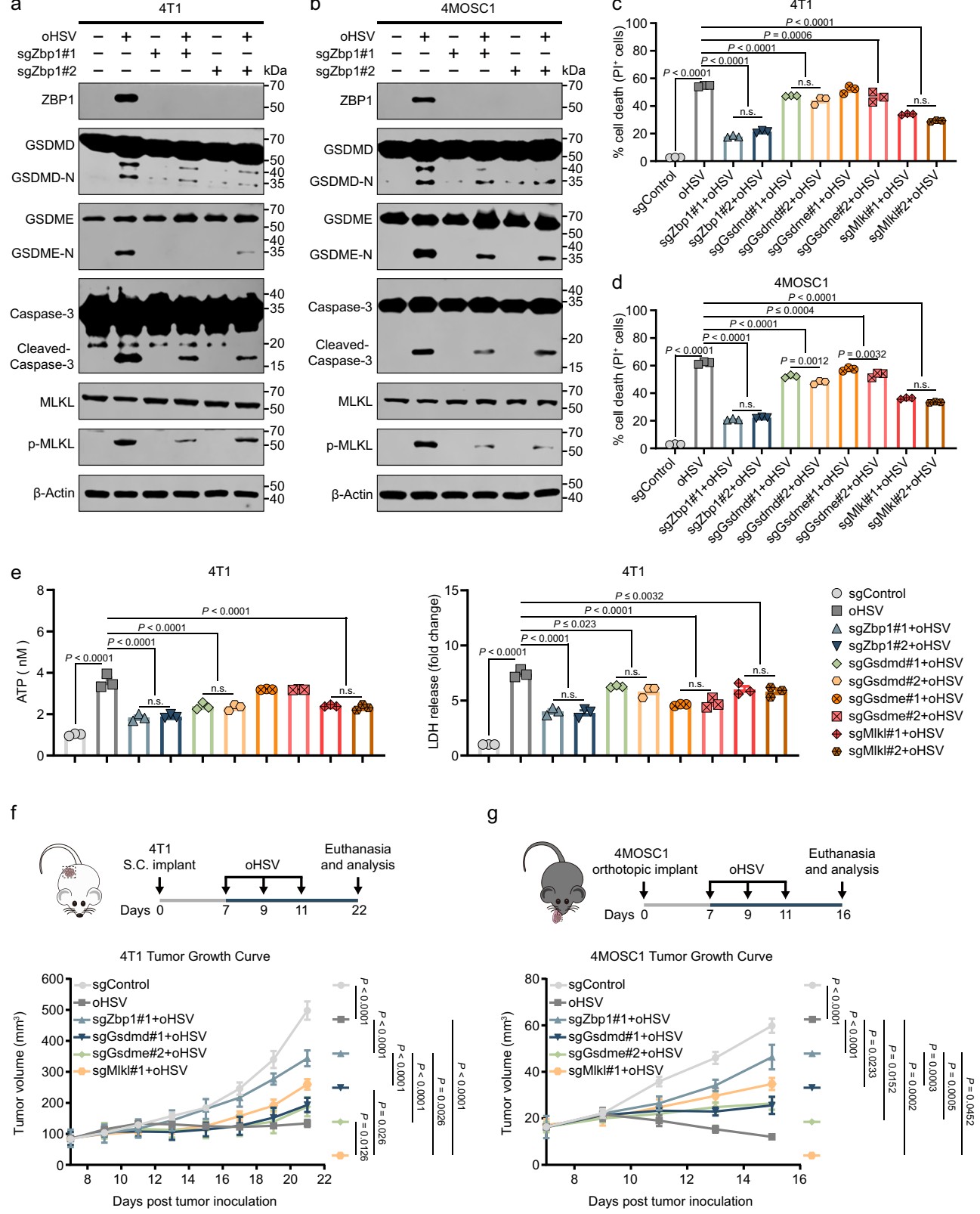

**Fig. 3 | oHSV induces ZBP1-mediated PANoptosis in tumor cells.** 4T1 (**a**) and 4MOSC1 (**b**) tumor cells were infected with oHSV (MOI = 1) for 24 h, cellular protein was analyzed by western blotting to detect the effect of ZBP1 expression on PANoptosis-related executive proteins. Effect of ZBP1, GSDMD, GSDME, and MLKL expression on relative cell viability in the 4T1 (**c**) and 4MOSC1 (**d**) cells treated with oHSV (MOI = 1) for 24 h. Western blotting was done thrice independently with similar results. n.s. not significant. **e** Effect of ZBP1, GSDMD, GSDME, and MLKL expression on ATP (left) and LDH (right) release in 4T1 tumor cells after oHSV (MOI = 1, 24 h)

treatment. *n* = 3 independent experiments. **f** Schematic illustration of injection of 4T1 tumor cells and the effect of ZBP1, GSDMD, GSDME, and MLKL expression on 4T1 tumor size after oHSV treatment. *n* = 5 mice. S.C. = Subcutaneous. **g** Schematic illustration of injection of 4T1 tumor cells and the effect of ZBP1, GSDMD, GSDME and MLKL expression on 4MOSC1 tumor size after oHSV treatment. *n* = 5 mice. Statistical significance was determined using two-tailed Student's *t* test in (**c**–**e**) and two-way ANOVA with Tukey's multiple comparisons test in (**f**, **g**). Data represent the mean ± s.e.m. Source data are provided in the Source Data file.

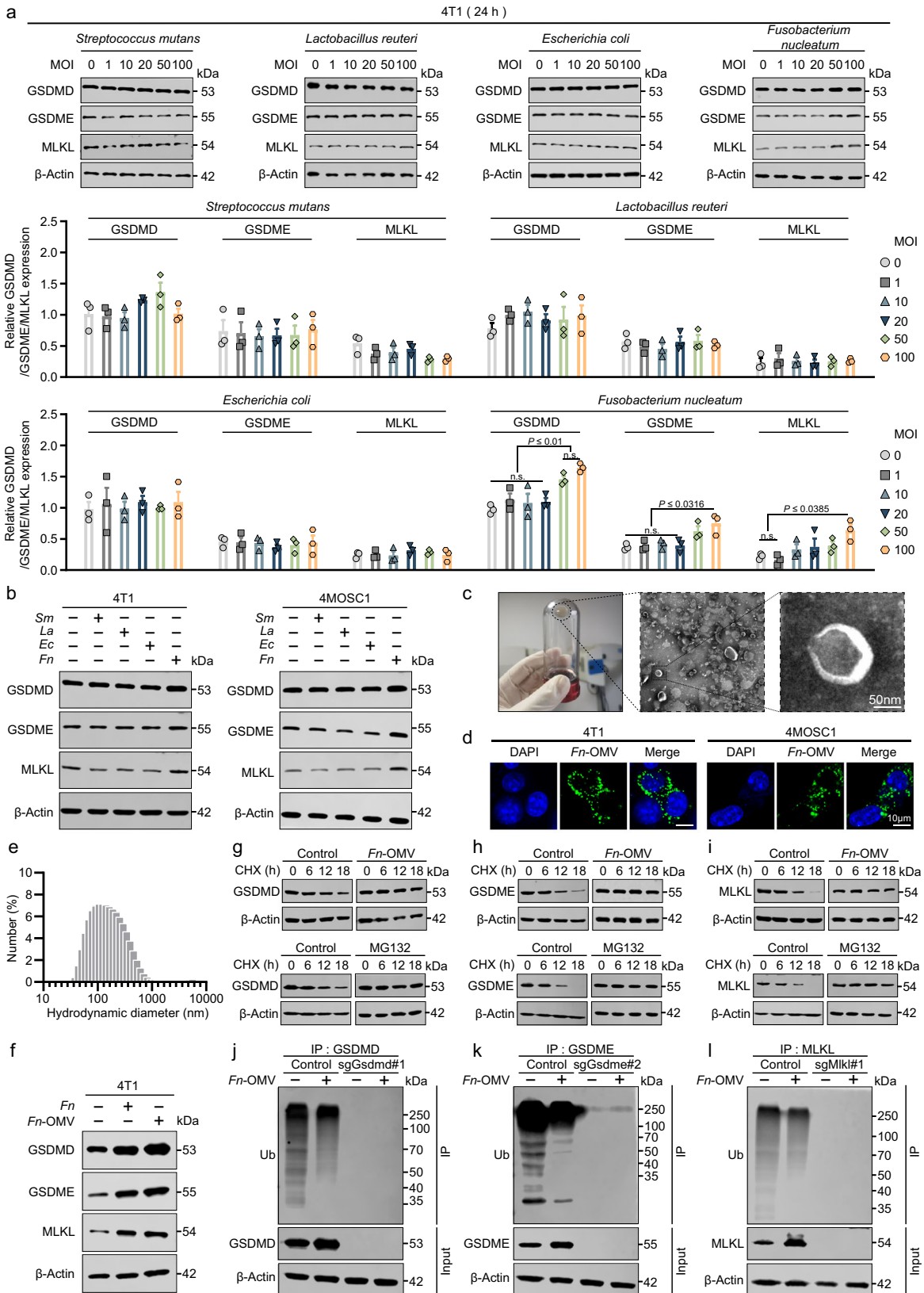

showed that compared to the groups treated with *Fn*-OMV alone or oHSV alone, the combination therapy of *Fn*-OMV and oHSV had the most optimal antitumor effect (Fig. 6a, b and Supplementary Fig. 5b). In addition, the combination therapy of *Fn*-OMV and oHSV induced the most significant inflammatory gene signature compared to the groups treated with *Fn*-OMV alone or oHSV alone (Fig. 6c). Furthermore, we

studied the immune cell profiles in tumor tissue and tumor-draining lymph nodes (TDLNs) after various treatments to elucidate the impact of the combination therapy of *Fn*-OMV and oHSV on antitumor immunity. The gating strategy of flow cytometry is shown in Supplementary Fig. S6. Mature DCs were induced to activate T cells by expressing CD80 and CD86 as co-stimulatory molecules. We observed

**Fig. 4 | *Fn*-OMV enhances the expression levels of PANoptosis effector proteins.**
**a** 4T1 cells were infected with different bacterial titers (MOI = 0, 1, 10, 20, 50, 100) for 24 h, cellular protein was analyzed by western blotting to detect the expression level of GSDMD, GSDME, and MLKL (upper) and corresponding quantitative analyses (below). Western blotting was done thrice independently with similar results. n.s. not significant. **b** The protein expression of GSDMD, GSDME, and MLKL was evaluated using western blotting in 4T1 (left) and 4MOSC1 (right) cells treated with different bacterial species. Western blotting was done thrice independently with similar results. n.s. not significant. **c** Schematic illustration of outer membrane vesicle (OMV) isolation and transmission electron microscopy image of *Fn*-OMV. Scale bars = 50 μm. Data were repeated thrice independently with similar results. **d** Confocal Laser Scanning Microscope shows the phagocytosis of *Fn*-OMV by 4T1 cells and 4MOSC1 cells. Scale bars = 10 μm. Data were repeated thrice independently with similar results. **e** Hydrodynamic diameter (Dh) of *Fn*-OMV in PBS at pH 7.4. Data were repeated thrice independently with similar results. **f** Effect of *Fn* and its secreted OMV on protein expression of GSDMD, GSDME, and MLKL was evaluated using western blotting in 4T1 cells. Western blotting was done thrice independently with similar results. n.s. not significant. Western blotting analysis was performed to evaluate the protein stability in 4T1 cells of GSDMD (**g**), GSDME (**h**), and MLKL (**i**) upon treatment with *Fn*-OMV (1 μg ml⁻¹) and MG132 (10 μM) in the presence of 50 μg ml⁻¹ cycloheximide (CHX). Immunoprecipitation (IP) and western blotting analysis were performed to examine the ubiquitination and expression of GSDMD (**j**), GSDME (**k**), and MLKL (**l**) in Control and Gsdmd, Gsdme, or Mlkl-depleted 4T1 cells after 24 h of *Fn*-OMV (1 μg ml⁻¹) treatment. Western blotting was done thrice independently with similar results. Statistical significance was determined using one-way ANOVA with Tukey's multiple comparisons test in (**a**). Data represent the mean ± s.e.m. Source data are provided in the Source Data file.

that in TDLNs, the combination therapy of *Fn*-OMV and oHSV significantly increased the fraction of mature DCs compared to the control group, *Fn*-OMV alone, or oHSV alone (Fig. 6d and Supplementary Fig. 7). In tumor tissue, the infiltration of CD8⁺ T cells increased most significantly in the combination therapy group of *Fn*-OMV and oHSV compared to the control group, *Fn*-OMV alone, or oHSV alone (Fig. 6e).

We also focused on four genes associated with T cell activation (interferon-γ, CD8α, granzyme B, and perforin 1) in gene expression profiling and found a correlation between T cell activation gene expression and treatment efficacy (Fig. 6f). Consistently, immunohistochemistry results showed a significant increase in CD8⁺ T cell infiltration in the combination therapy group of *Fn*-OMV and oHSV compared to the control group, *Fn*-OMV alone, or oHSV alone (Fig. 6g). Importantly, in tumor tissue, the combination therapy of *Fn*-OMV and oHSV effectively converted M2 tumor-associated macrophages (TAMs) into M1 phenotype (Fig. 6h and Supplementary Fig. 8a, b). In TDLNs and tumor tissue, the combination therapy of *Fn*-OMV and oHSV resulted in the most significant reduction in the number of regulatory T cells (Tregs) compared to the control group, *Fn*-OMV alone, or oHSV alone (Fig. 6i, j and Supplementary Fig. 9). In addition, we evaluated the proliferation of tumor tissue using Ki-67 staining. The expression of Ki-67 was significantly reduced in the combination therapy group of *Fn*-OMV and oHSV compared to the control group, *Fn*-OMV alone, or oHSV alone (Supplementary Fig. 10). We also investigated the in vivo biosafety of intra-tumoral injection of oHSV and *Fn*-OMV. Histological examination (H&E staining) of the heart, liver, spleen, lungs, and kidneys of the treated mice showed no abnormal cellular morphology compared to the control mice (Supplementary Fig. 11 and Supplementary Fig. 12). In addition, routine blood tests and blood biochemistry analysis revealed no significant differences in various indicators compared to the control mice (Supplementary Fig. 13). Based on observed post-treatment increases in CD8⁺ T cell infiltration, the upregulation of inflammatory markers in the tumor microenvironment, and an enhanced response to ICB, as shown in Fig. 7g, we hypothesize that treated tumors demonstrate features consistent with immunogenicity or "hot" tumors (Fig. 6k).

### The combination of *Fn*-OMV and oHSV enhances the effectiveness of anti-PD-L1 immunotherapy and stimulates immune memory

The low response rate of anti-PD-1/PD-L1 immunotherapy is a major bottleneck in antitumor immunity[45]. Insufficient infiltration of cytotoxic T lymphocytes and loss of PD-L1 expression may be underlying factors for resistance in PD-1/PD-L1-directed ICB. Through mIHC, we visually observed an increase in CD8⁺ T cell infiltration, a decrease in Treg cells, and a significant upregulation of PD-L1 expression in tumor tissue treated with the combination therapy of *Fn*-OMV and oHSV (Fig. 7a, b). Flow cytometry results further confirmed the upregulation of PD-L1 expression in tumor cells by the combination therapy of *Fn*-OMV and oHSV (Fig. 7c). These findings suggest that the combination

therapy of *Fn*-OMV and oHSV has the potential to improve the efficacy of ICB treatment. Furthermore, we observed a significant increase in the number of central memory T cells (Tcm) expressing CD44⁺CD62L⁺ in the TDLNs of the combination therapy group of *Fn*-OMV and oHSV compared to the control group, *Fn*-OMV alone, or oHSV alone (Fig. 7d and Supplementary Fig. 14). Further in vivo experimental results demonstrated that the combination therapy of *Fn*-OMV, oHSV, and anti-PD-L1 (10 mg kg⁻¹; BE0101; Bio X Cell) had better antitumor efficacy (Fig. 7e, f) and longer survival (Fig. 7g) compared to the combination therapy of *Fn*-OMV and oHSV alone. We conducted in vivo CD8⁺ T cell depletion in mice using CD8α monoclonal antibody (10 mg kg⁻¹; BE0004-1; Bio X Cell) to further validate the role of CD8⁺ T cells in tumor rejection observed in Fig. 7f. The experimental results demonstrate that CD8⁺ T cells indeed mediate tumor rejection responses (Supplementary Fig. 15a, b). We performed repeat tumor challenge experiments in cured mice and naive mice to further demonstrate the long-term protective effects of treatment-induced memory T lymphocytes against tumors in mice. The experimental results suggested that treatment-induced memory T lymphocytes confer long-term protection against murine tumors (Supplementary Fig. 16a–c). Finally, upon injection of the primary flank tumor, we injected a second flank tumor to demonstrate abscopal activity. In the treatment group receiving oHSV + *Fn*-OMV + PD-L1, abscopal activity was observed with the rejection of a second flank tumor following injection of the primary flank tumor (Supplementary Fig. 17a–c). We conducted additional experiments combining our oHSV/*Fn*-OMV treatment with CTLA-4 blockade (10 mg kg⁻¹; BE0164; Bio X Cell). The results demonstrated good synergistic effects when oHSV/*Fn*-OMV treatment was combined with CTLA-4 blockade (Supplementary Fig. 18a–c).

In addition, the lung metastasis was significantly more severe in the control group and the group treated with anti-PD-L1 alone compared to the group treated with the combination therapy of *Fn*-OMV and oHSV, as well as the group treated with the combination therapy of *Fn*-OMV, oHSV, and anti-PD-L1 (Fig. 7h, i). However, the reduction in lung metastases could potentially result from the destruction of the primary tumor by direct inoculation rather than a direct effect on metastasis. To rule out the influence of primary tumor burden, we designed additional lung metastasis experiments.

Specifically, we injected non-luciferase 4T1 cells subcutaneously in the back, then performed intravenous injection of luciferase-expressing 4T1-luc cells one day after treatment completion. Bioluminescence imaging was used to assess the impact of treatment on lung metastasis. The results demonstrated treatment could indeed reduce lung metastasis (Supplementary Fig. 19a, b).

To provide a visual representation of the main findings in this study, we have created an illustrative diagram (Fig. 8).

### Discussion
OVs have emerged as a promising anticancer modality, albeit with recognized constraints as monotherapies[1]. Addressing the need to

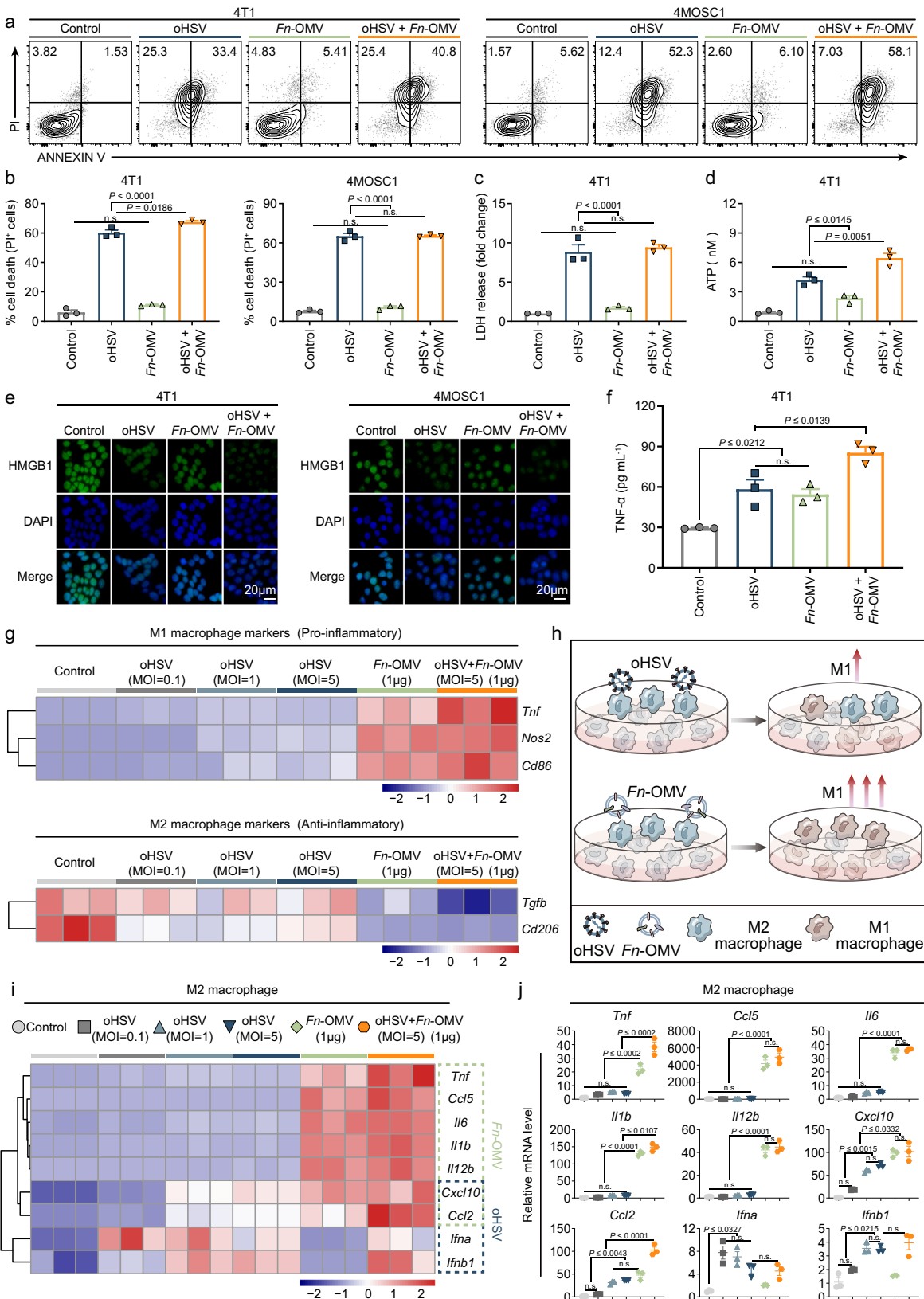

augment their lytic efficacy is therefore of paramount importance. Our findings suggest that oHSV-induced interferon response within tumor cells facilitates ZBP1-driven PANoptosis by elevating Z-RNA levels. Based on these findings and our previous research results, we screened and identified *Fn*-OMVs that can enhance the expression levels of PANoptosis execution proteins GSDMD/E and MLKL. This combination

strategy, encompassing *Fn*-OMVs and oHSV, bolstered inflammatory responses and exhibited profound antitumor immune activation in vivo, transforming immunologically "cold" tumors "hot" and heightening the effectiveness of ICB.

The innate immune sensors serve as vital components of the innate immune system, responsible for sensing intracellular pathogen

**Fig. 5 | Fn-OMV and oHSV have great potential for synergistic application.**
Evaluation of cell apoptosis by flow cytometry at 24 h in different treatment groups. Representative flow cytometric plots (**a**) and relative cell viability (**b**) of different treatment groups are shown in the 4T1 (left) and 4MOSC1 (right) cells. The release of LDH (**c**) and ATP (**d**) from 4T1 cells after oHSV (MOI = 1) treatment and Fn-OMV (1 μg ml⁻¹) alone or in combination. **e** Immunofluorescence detection of HMGB1 expressed on 4T1 (left) and 4MOSC1 (right) cells was observed using a Confocal Laser Scanning Microscope. Scale bars, 20 μm. **f** 4T1 cell culture supernatants were assayed for TNF-α by ELISA after oHSV (MOI = 1) treatment and Fn-OMV (1 μg ml⁻¹) alone or in combination. **g** Heatmap showing the expression of signature genes related to mRNA expression of M1 macrophage markers (upper) and M2 macrophage markers (below) in oHSV treatment and Fn-OMV alone or in combination. **h** Schematic illustration shows the effect of oHSV and Fn-OMV on the polarization regulation ability of macrophages. Heatmap (**i**) and bar graphs (**j**) showing relative mRNA levels of inflammatory-associated genes in M2 macrophages following treatment with oHSV and Fn-OMV alone or in combination. $n$ = 3 independent experiments for **b**–**d**, **f**, **g**, **i**, **j**. Statistical significance was calculated via one-way ANOVA with Tukey's multiple comparisons test. n.s. not significant. Data represent the mean ± s.e.m. Source data are provided as a Source Data file.

invasion and inducing cell death in infected cells[46], playing a critical role in antitumor immunity[47]. It is known that HSV-1 activates the absent in melanoma 2 (AIM2) PANoptosome with ZBP1 and Pyrin in bone marrow-derived macrophages (BMDMs)[19]. Our findings suggest that during HSV-1 infection, AIM2 expression is upregulated in some tumor cell lines (4MOSC1and 4MOSC2) but downregulated in others (SCC7, 4T1, and CT26), while ZBP1 expression is upregulated across all five tumor cell lines tested. Further investigation revealed that ZBP1-mediated PANoptosis plays an important role in the antitumor effects of oHSV in cancer. These findings indicate that innate sensors may be modulated differently during oHSV infection in tumor cells, as compared to their regulation in macrophages previously reported. We speculated several possibilities for the discrepancies: differential expression of regulatory proteins that balance AIM2 activity and maintain homeostasis[48]; viral evasion tactics that suppress AIM2 to avoid immune detection[49]; and the heterogeneous cellular and immune contexts within the tumor microenvironment that influence AIM2's roles[50]. These dynamics could be attributed to the observed downregulation of AIM2 in certain cancer scenarios. Previous studies have found that virus intracellular replication such as influenza virus (IAV) and vaccinia virus (VACV) can generate a substantial amount of Z-RNA to activate ZBP1[37,51], but it also been observed that intracellular replication of viruses like rhabdovirus (VSV) or poliovirus (EMCV) does not produce detectable Z-RNA[37]. In this study, we have identified that oHSV primarily activates ZBP1 in tumor cells through the production of endogenous Z-RNA.

Our study demonstrates that oHSV infection of tumor cells concurrently promotes pyroptosis, apoptosis, and necroptosis. This cell death effect is significantly impeded upon Zbp1 knockout, leading to a marked reduction in the antitumor efficacy of oHSV. These findings underscore the critical role of ZBP1-mediated PANoptosis in eliciting antitumor responses. While apoptosis is generally considered an immunologically inert form of cell death that lacks the release of cellular contents[52], lytic cell death, such as pyroptosis and necroptosis, provokes the release of immunogenic substances[53]. In this study, stimulation with oHSV led to an increase in ATP levels; however, it is noteworthy that the enhanced production of ATP may not always confer beneficial antitumor immunity, as ectoenzymes CD39 and CD73 can metabolize extracellular ATP into adenosine (ADO), exerting immunosuppressive effects. The process of immune stimulation is invariably accompanied by negative feedback mechanisms. For instance, treatment with our oHSV and Fn-OMV approach results in the upregulation of inhibitory immune checkpoint molecules such as PD-L1 and CTLA-4. Future research should further investigate the impact of ATP and ADO on overall therapeutic outcomes.

Both heat-killed bacteria and OMVs possess distinct advantages and research applications. OMVs production is generally costly and faces scalability challenges in mass production. In contrast, heat-killed bacteria are readily amenable to large-scale preparation. Heat-killed bacteria can serve as vaccines for immunological research or as experimental models to elicit host immune responses due to their preserved antigens. OMVs, containing a variety of bacterial components, are often used in vaccine design or adjuvants for studying bacterial-host cell interactions, as well as in drug delivery systems.

OMVs are typically considered safer as they do not encompass complete bacterial genomes[54]. Given the current controversy surrounding Fn role in tumorigenesis and its association with certain tumors as a potential risk factor, we cautiously chose to investigate Fn-OMVs from a safety perspective. Comparative analysis of the pros and cons of heat-killed Fn and Fn-OMVs necessitates extensive future work.

The use of the Fn-OMV is a nice addition to the therapy. However, it is not a defined substance so quality control for patient use may prove challenging. In the future, the active ingredients will likely require definition. A potential limitation of the current approach is the need for direct intratumoral injection to achieve localized therapeutic effects. For tumors that are difficult to access surgically or tumors with widespread metastatic disease, this requirement for direct administration may limit applicability. Further development is needed to optimize delivery methods that could enable broader application to additional cancer types and disease stages.

In summary, this study reveals that oHSV likely activates ZBP1 in tumor cells primarily through the accumulation of endogenous Z-RNA. Mechanistically, oHSV induces the production of a large number of ISGs, and the elevation of ISGs containing Z-prone regions within 3′-UTR is a significant source of intracellular Z-RNA. ZBP1 is sensitive to the presence of Z-conformations in nucleic acids, thus detecting these specific Z-structures within the ISGs' 3′-UTRs, leading to its upregulation. We have also discovered that oHSV-induced ZBP1-mediated PANoptosis plays a significant role in antitumor effects. Based on these findings, the combined treatment using Fn-OMVs and oHSV has effectively suppressed tumor growth and improved the efficacy of ICB, providing a strategy for cancer treatment.

## Methods

### Ethical Statements

Ethical approval for this study was granted by the Animal Ethics Committee of the School and Hospital of Stomatology of Wuhan University (approval number: S07922090D). All animal experimental procedures adhered to the Regulations for the Administration of Affairs Concerning Experimental Animals approved by the State Council of the People's Republic of China. The mice were housed under specific pathogen-free (SPF) conditions with a 12/12-h light/dark cycle, temperature of approximately 22 °C, and humidity of around 50%.

### Cell lines, animal models and treatments

The murine breast cancer cell line 4T1 (Cat CRL-2539), obtained from the American Type Culture Collection (ATCC), was cultured in RPMI 1640 medium supplemented with 10% FBS (Gibco) and 1% penicillin (HyClone). The murine colon carcinoma cell line CT26 (Cat TCM37), purchased from the Shanghai Cell Bank of the Chinese Academy of Sciences, was cultured in DMEM medium supplemented with 10% FBS and 1% penicillin. The 4MOSC1 and 4MOSC2 cells were gifts from Prof. J. Silvio Gutkind at the University of California San Diego via a material transfer agreement (SD2017-202), and these cells were cultured in keratinocyte serum-free medium (K-SFM, Gibco-BRL). The mouse SCC7 head and neck squamous cell carcinoma cell line (gifted by Prof. Qianming Chen; West China School and Hospital of Stomatology

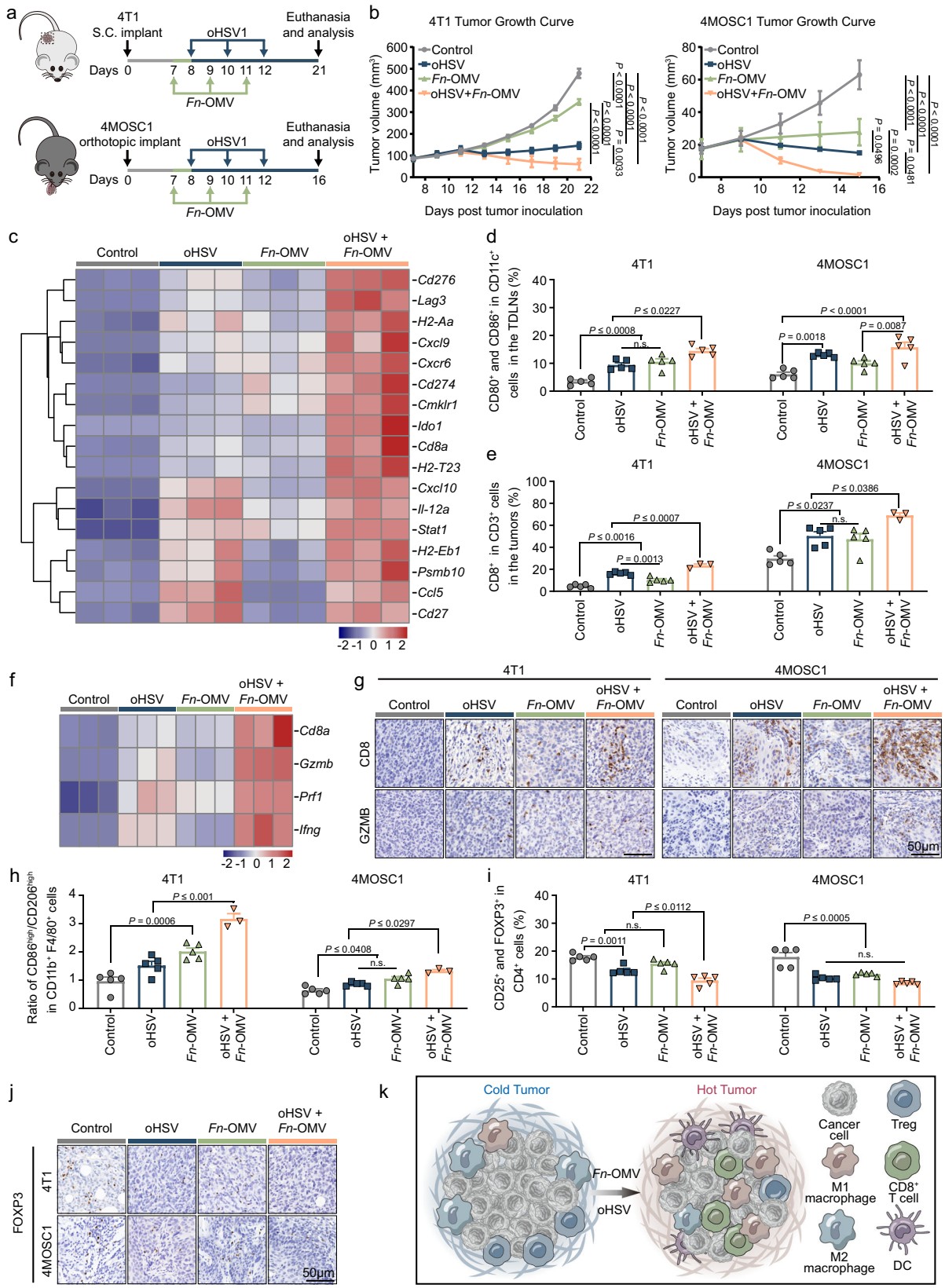

Sichuan University, Chengdu, China) was cultured in RPMI 1640 medium with 10% FBS and 1% penicillin. RAW 264.7 (Cat TIB-71) obtained from the ATCC was cultured in DMEM medium supplemented with 10% FBS and 1% penicillin.

Female BALB/c (Figs. 1d, 3f, 6a, 7e and Supplementary Figs. 15–19), male C57BL/6 (Supplementary Fig. 5b) and female C57BL/6 mice (Figs. 3g and 6a), aged 6–8 weeks and weighing 18–20 g, were obtained from Hubei Provincial Academy of Preventive Medicine. 4T1 cells

**Fig. 6 | Combination of *Fn*-OMV and oHSV converts "cold" tumors into "hot" tumors. a** Schematic representation of 4T1 (upper) and 4MOSC1 (below) tumor inoculation and treatment with oHSV and *Fn*-OMV. S.C. = Subcutaneous. **b** Average 4T1 (left) and 4MOSC1 (right) tumor growth curve depicted as mean ± s.e.m. for each treatment group (*n* = 5 mice). Statistical significance was evaluated using two-way ANOVA. **c** Heatmap shows a 17-gene expression profile associated with inflammation produced in mice of different treatment groups. **d** Immunological profile of TDLNs. This includes the frequency of the lymph nodes CD80⁺ CD86⁺ population in CD11c⁺ cells in 4T1 (left) and 4MOSC1 (right) tumors. **e** CD8⁺ T cell populations in the tumors were measured using flow cytometry, and quantitative analyses were performed in 4T1 (left) and 4MOSC1 (right) tumors. **f** Heatmap shows the expression signature of four selected genes, all highly associated with CD8⁺ T cell activation, across different treatment groups. **g** Representative images of immunohistochemical analysis of CD8 and Granzyme B (GZMB) expression in different treatment groups of 4T1 tumors (left)

and 4MOSC1 tumors (right), Scale bars, 50 μm. The images of immunostaining (**g**) were representative of those generated from five mice in each group. **h** CD86⁺ macrophages and CD206⁺ macrophages in the tumors were measured using flow cytometry, and quantitative analyses were performed. **i** CD25⁺ and FOXP3⁺ Treg cell populations in the TDLNs were measured using flow cytometry, and quantitative analyses were performed. **j** Representative images of immunohistochemical analysis of FOXP3 expression in different treatment groups of 4T1 tumors (upper) and 4MOSC1 tumors (below), Scale bars, 50 μm. The images of immunostaining (**j**) were representative of those generated from five mice in each group. **k** A schematic diagram illustrating the transformation of "cold" tumors into "hot" tumors by the combined treatment of oHSV and *Fn*-OMV. All data are shown as the mean ± s.e.m. *n* = 3-5 independent experiments for (**c–f**, **h**, **i**). Statistical significance was calculated via one-way ANOVA with Tukey's multiple comparisons test in (**d**, **e**, **h**, **i**). n.s. not significant. Source data are provided as a Source Data file.

(1 × 10⁶) were injected subcutaneously into the right hind flank of each BALB/c mouse, while 4MOSC1 cells (1 × 10⁶) were transplanted into the dorsum linguae of each C57BL/6 mouse.

Once the tumor volume reached approximately 100 mm³ for 4T1 or 20 mm³ for 4MOSC1, the mice were administered oHSV (4T1: 2 × 10⁶ pfu; 4MOSC1: 0.4 × 10⁶ pfu) or *Fn*-OMV (4T1: 5 μg; 4MOSC1: 1 μg) via intratumoral injection. The injections were repeated every other day for a total of three times. In addition, the mice were given intraperitoneal injections of PD-L1 antibody (10 mg kg⁻¹; BE0101; Bio X Cell), CD8 antibody (10 mg kg⁻¹; BE0004-1; Bio X Cell), and CTLA-4 antibody (10 mg kg⁻¹; BE0164; Bio X Cell) every three days after tumor formation. Tumor size was monitored every two days, and strict observation of the mice was conducted throughout the experiment. At the conclusion of the treatment, the mice were euthanized, and the tumors were dissected for further analysis. All mice in this study were euthanized under CO₂ anesthesia if the volume of the primary tumor reached a maximum allowable volume of 1500 mm³ (4T1) or 150 mm³ (4MOSC1) or if the tumor burden compromised the animal welfare. The maximal tumor size in this study was not exceeded.

To evaluate the efficacy of oHSV, *Fn*-OMV, and αPD-L1 for tumor-infiltrating CD8⁺ T cells (shown in Supplementary Fig. 15), 1 × 10⁶ 4T1 cells were injected at the indicated site. Tumor-bearing mice were received three anti-CD8 injections of depleting antibodies for CD8⁺ T cells and treated with oHSV + *Fn*-OMV + αPD-L1 three times.

For the tumor re-challenge experiments (shown in Supplementary Fig. 16), transplanted 4T1 tumors were treated with oHSV + *Fn*-OMV + αPD-L1 three times. Fifty days after the final treatment, five mice that responded completely to the combination treatment or five naïve mice were selected to be rechallenged by subcutaneous injection of 1 × 10⁶ 4T1-Luc cells into the contralateral back. Bioluminescence imaging of mice was performed in days 80.

In experiments where rechallenge was conducted in setting of a primary tumor (shown in Supplementary Fig. 17), 1 × 10⁶ 4T1 cells were inoculated on day 0, and 1 × 10⁶ 4T1 cells were inoculated on the contralateral back two days later. For bilateral subcutaneous tumor treatment models, one tumor was treated and both the treated tumor and the untreated tumor were assessed for growth. To evaluate the efficacy of oHSV + *Fn*-OMV on αCTLA-4 (shown in Supplementary Fig. 18), 1 × 10⁶ 4T1 cells were injected at the indicated site. Mice were checked every two days to examine the sizes of primary tumors. Mantel-Cox analysis of Kaplan-Meier curves was performed to analyze statistical differences in overall survival time with a general tumor size cut-off of 1500 mm³.

To investigate the combined therapeutic efficacy of oHSV + *Fn*-OMV + αPD-L1 in controlling lung metastatic colonization by 4T1-Luc cells, BALB/c mice were implanted with 1 × 10⁶ 4T1 cells on the back (shown in Supplementary Fig. 19). After three treatments, 5 × 10⁵ 4T1-Luc cells were injected into the mice through the tail vein. At the endpoint (days 30), pulmonary metastasis was detected by bioluminescence imaging with an IVIS in vivo imaging system (Lumina Series III, PerkinElmer, USA).

## Flow cytometry

Tumors and tumor-draining lymph nodes (TDLNs) obtained from the experimental mice were processed into a single-cell suspension using the gentleMACS Dissociator from Miltenyi Biotec. The cells were then stained with antibodies and analyzed. FlowJo 10 software (Version 10.0.6, Tree Star) was utilized to analyze and visualize the research results. For fluorescence labeling, the antibodies used included Fixable Viability Dye (1:1000; eFluor 506; eBioscience), anti-CD45 (1:500; APC-CY7; eBiosciences), anti-CD3 (1:500; FITC; eBiosciences), anti-CD4 (1:500; eFluor450; eBiosciences), anti-CD8 (1:500; PerCP-Cyanine 5.5; BioLegend), anti-CD11b (1:500; FITC; Biolegend), anti-CD11c (1:500; FITC; eBiosciences), anti-CD80 (1:300; PE; eBiosciences), anti-CD86 (1:300; APC; eBiosciences), anti-MHC-II (1:300; PE-Cyanine7; eBioscience), anti-CD25 (1:200; APC; eBioscience), anti-FOXP3 (1:100; PE; eBioscience), anti-CD44 (1:500; PE; eBiosciences), anti-CD62L (1:200; APC; eBiosciences), anti-F4/80 (1:200; PE-Cyanine7; eBioscience), anti-CD206 (1:200; APC; BioLegend), anti-CD86 (1:200; PE; BioLegend), and anti-PD-L1 (1:200; PE; Biolegend) following the manufacturers' instructions. Apoptosis was detected using the Annexin V-FITC/PI Apoptosis Kit from MULTI SCIENCES. FITC and PI detection channels on the Beckman CytoFLEX flow cytometer (Beckman Coulter) were used to detect Annexin V-FITC and PI.

## ATP detection

4T1 or 4MOSC1 cells were seeded into 24-well plates at a density of 2 × 10⁵ cells per well and cultured. After the cells were attached to the plate, the medium was replaced with a serum-free medium, and the corresponding treatment was applied. After 24 h of stimulation, the supernatant was collected for further analysis. ATP release in each group was measured using an ATP detection kit (Beyotime).

## LDH detection

4T1 cells were seeded in a 96-well plate at a density of 5 × 10³ cells per well and incubated for 12 h. After the cells were attached to the plate, fresh medium without FBS was added, and the corresponding treatment was applied. The plate was then incubated at 37 °C for 24 h, with untreated cells serving as the control group. To measure LDH release, the supernatant was collected from each well after centrifugation at 400 × *g* for 5 min, and 120 μL of the supernatant was transferred to a new 96-well plate. Next, 60 μL of LDH detection working solution (Beyotime) was added to the 96-well plate and incubated at 37 °C for 30 min. Finally, the absorbance was measured at 490 nm using a microplate reader.

## Bacteria culture

*Fusobacterium nucleatum* (*Fn*) ATCC 25586 (GDMCC VPI 4355, Guangzhou, China) and *Streptococcus mutans* (*Sm*) ATCC 700610 (GDMCC UA 159, Guangzhou, China) were cultured in brain heart infusion (BHI) broth (BD Biosciences, USA). *Lactobacillus reuteri* (*La*) strain ATCC 23272 (CCTCC AB 2014289, Wuhan, China) was cultured

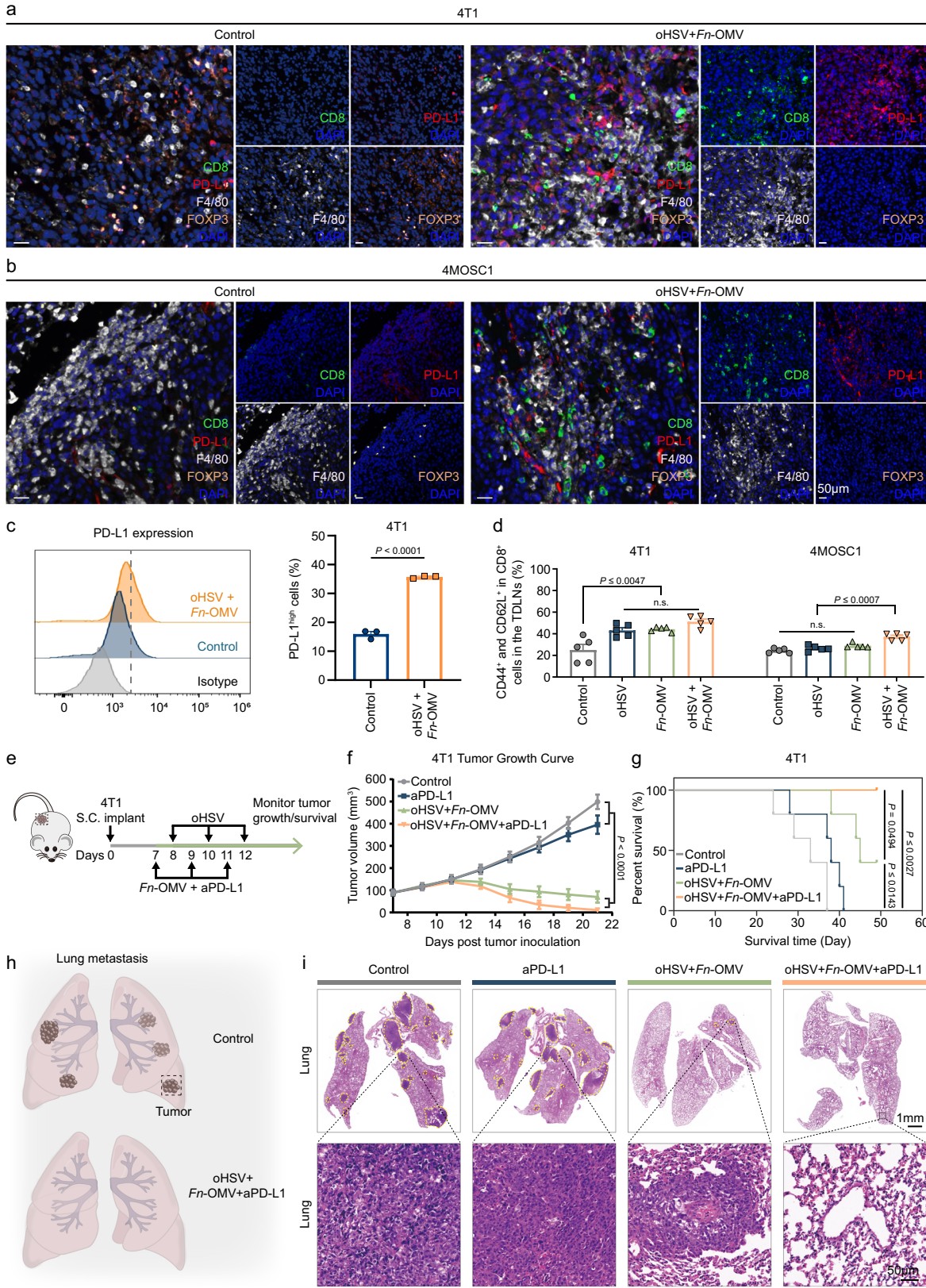

in Man Rogosa Sharp (MRS) medium (BD Biosciences, USA). *Escherichia coli* (*Ec*) ATCC 8739 (CCTCC AB 2016285) was cultured in Luria Broth medium (Land Bridge, Beijing). The bacterial cultures were maintained in an anaerobic chamber (Mart Microbiology, Netherlands) under an atmosphere of 90% $N_2$, 5% $CO_2$, and 5% $H_2$ at a temperature of 37 °C.

**Preparation and characterization of Bacteria-OMV**
To isolate bacteria-OMV, the bacteria were cultured in broth and subsequently collected by centrifugation at 10,000 × *g* for 30 min at 4 °C. The resulting culture supernatant was filtered using a 0.22 μm filter (Millipore) and then concentrated 10-fold through centrifugation at 10,000 × *g* at 4 °C using Amicon Ultra Centrifugal Filters (100 kDa;

**Fig. 7 | Combination of *Fn*-OMV and oHSV enhances the effectiveness of anti-PD-L1 immunotherapy and stimulates immune memory.** The expression of CD8 (green), PD-L1 (red), F4/80 (white), and FOXP3 (orange) was observed in 4T1 tumors (**a**) and 4MOSC1 tumors (**b**) using multiplex immunohistochemistry (mIHC). Scale bars, 50 μm. The images of mIHC (**a**, **b**) were representative of those generated from five mice in each group. **c** PD-L1 expression on 4T1 cells following combination treatment of oHSV and *Fn*-OMV was observed using flow cytometry (*n* = 3 independent experiments). Statistical significance was determined using a two-tailed Student's *t* test. **d** CD44+ and CD62L+ T cell populations in the TDLNs were measured using flow cytometry, and quantitative analyses were performed (*n* = 5 independent experiments). Statistical significance was calculated via one-way ANOVA with Tukey's multiple comparisons test. **e** Experimental schedule for 4T1

tumor inoculation and treatment with oHSV, *Fn*-OMV, and aPD-L1. **f** Tumor volume of 4T1 tumor-bearing mice with different treatments (*n* = 5 mice). S.C. = Subcutaneous. Statistical significance was calculated via two-way ANOVA with Tukey's multiple comparisons test. **g** Survival analysis of 4T1 tumor-bearing mice after the different treatments (*n* = 5 mice). **h** Schematic of the experimental process for lung metastasis model. **i** Typical images of lung tissues, HE staining images of metastatic foci in lungs from the different treatment groups. Scale bars, 50 μm. The images of HE staining (**i**) were representative of those generated from five mice in each group. All data are shown as the mean ± s.e.m. Statistical significance was calculated via one-way ANOVA with Tukey's multiple comparisons test. n.s. not significant. Source data are provided as a Source Data file.

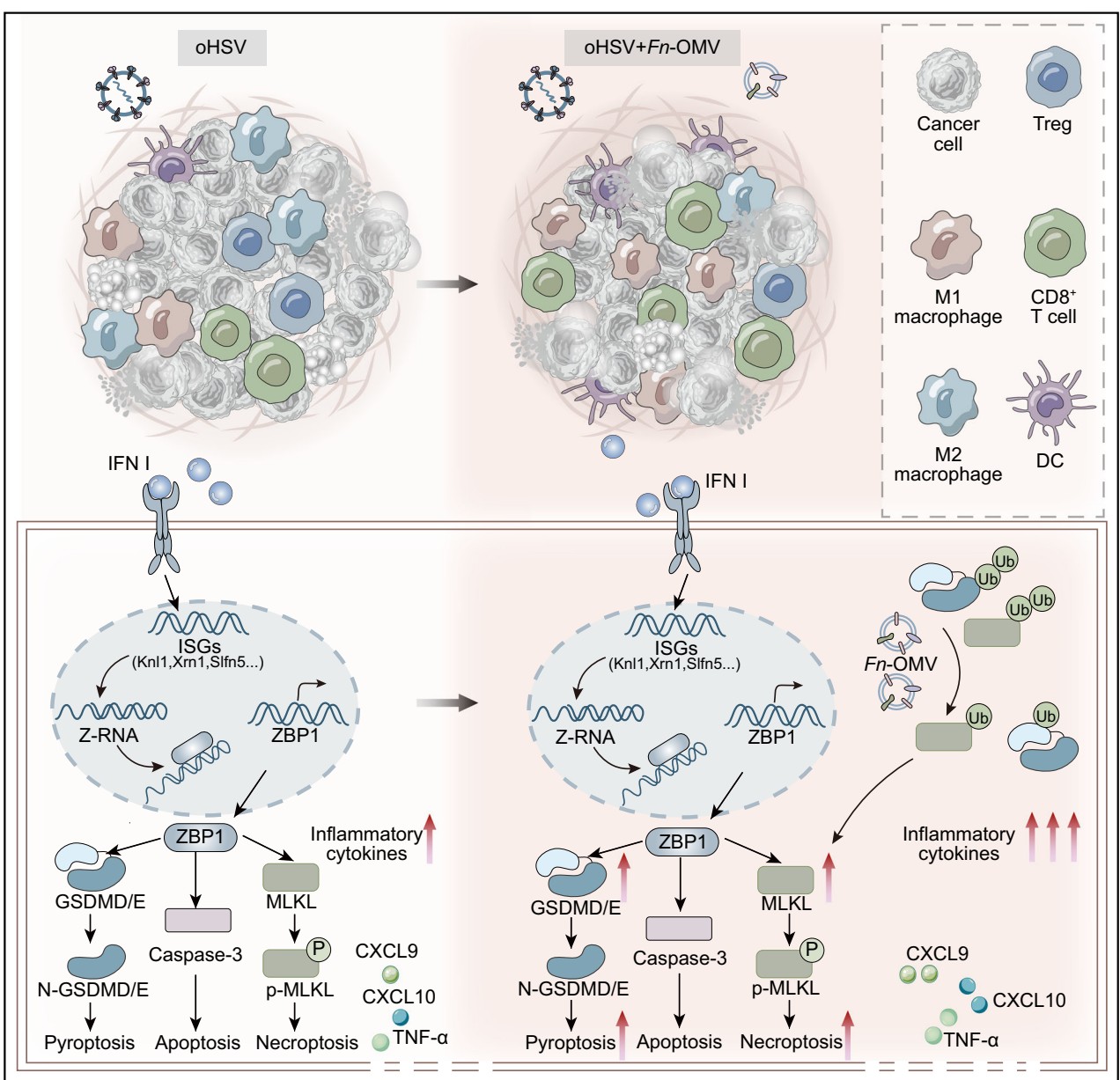

**Fig. 8 | Illustration of the Mechanism of Combined *Fn*-OMV and oHSV Therapy.** The diagram illustrates the process of oHSV-induced ZBP1-mediated PANoptosis, as well as the enhancement of oHSV-induced ZBP1-mediated PANoptosis by *Fn*-OMV.

Merck Millipore). Then concentrated solution underwent further centrifugation for 3 h at 100,000 × g and 4 °C to pellet the bacteria-OMV. The obtained bacteria-OMV were resuspended in PBS. The protein content of the bacteria-OMV was quantified using a BCA protein assay kit (Beyotime). The morphologies and diameters of the bacteria-OMV were assessed using a transmission electron microscope (TEM; JEM-2100, JEOL) and dynamic light scattering (DLS; Malvern, Zetasizer Nano ZS, UK). The bacteria-OMV were stored at −80 °C until downstream experiments.

### Generation of oncolytic HSV-1 Δ34.5/Δ47

ICP34.5- and ICP47-deficient HSV-1 (HSV-1 Δ34.5/Δ47; referred to as oHSV from here onwards) was created by deleting the coding sequences of ICP34.5 and ICP47 in a bacterial artificial chromosome (BAC) using a two-step Red-mediated recombination method[55].

### Quantitative real-time polymerase chain reaction (qRT–PCR)

For the detection of inflammasomes, select key innate sensor-related genes, and ISGs in tumor cells, we introduced oHSV with a viral titer of $1 \times 10^9$ TU/ml and an MOI of 1, and stimulated the cells for 24 h. To assess the transition of M1/M2 macrophage polarization, RAW 264.7 cells were treated with oHSV (MOI = 0.1, 1, 5) or Fn-OMV at a concentration of $1 \mu g \, ml^{-1}$ for 12 h. For the evaluation of inflammatory gene expression in tumor tissues, the tissues were homogenized using a cryo-grinding technique to obtain a consistent tissue lysate. The tumor tissues or cultured cells were subjected to total RNA extraction following the manufacturer's instructions (Axygen). The concentration of RNA was determined using a spectrometer (Shimadzu UV-2401PC). Reverse transcription of one microgram of total RNA was carried out using HiScript II Reverse Transcriptase (Vazyme Biotech), following the manufacturer's protocol. Approximately 1% of the resulting cDNA was utilized as a template in each RT-PCR reaction, performed with the SYBR master mix (Vazyme Biotech). The RT-PCR protocol employed three steps. The $2^{-\Delta\Delta CT}$ method was employed to calculate the fold induction, with each cDNA data point normalized to GAPDH expression. The primer sequences can be found in Supplementary Data 1. To effectively illustrate the qRT-PCR results, we utilized the heatmap function in R Studio software to plot the normalized gene expression data. The heatmap's color palette was strategically chosen to represent the continuum of expression levels: shades of blue denoted down-regulated gene expression compared to the control, while a progression towards red signified upregulated expression. This method produced a visually intuitive gradient, facilitating an immediate perception of gene expression alterations.

### Western blotting analysis and Enzyme-Linked Immunosorbent Assay (ELISA)

The protein concentration of the extracted sample was determined, and then the samples were loaded onto 5% SDS–PAGE concentrated gels for stacking and 12% SDS-PAGE separating gels for separation. Following electrophoresis, the proteins were transferred onto PVDF membranes and subsequently incubated with 5% skimmed milk at room temperature for 1 h, followed by overnight incubation with the primary antibody solution at 4 °C. The next day, the membranes were incubated with the appropriate secondary antibody for 1 h, and the Odyssey system (LI-COR Biosciences) was utilized to visualize the image using an ECL kit (Advansta)[56]. For protein quantification, ImageJ software (ImageJ, National Institutes of Health, Bethesda, Maryland, USA) was employed. The experiment was repeated at least three times. Primary antibodies against GSDMD (1:1000; ab219800; Abcam), GSDME (1:1000; ab215191; Abcam), MLKL (1:1000; 37705; Cell Signaling Technology), Phospho-MLKL (1:1000; 37333; Cell Signaling Technology), beta Actin (1:1000; ab8226; Abcam), Lamin A/C (1:2000; 4777; Cell Signaling Technology), Caspase-3 (1:1000; 9662; Cell Signaling Technology), Cleaved Caspase-3 (1:1000; 9661; Cell Signaling Technology),

ZBP1 (1:1000; AG-20B-0010; AdipoGen), NLRP6 (1:1000; A15628; ABclonal) and Ubiquitin (1:1000; 10201; Proteintech) were used. For ELISA, the cell culture supernatants were obtained. TNF-α concentrations in supernatants were measured using commercial ELISA kits (4 A Biotech) in accordance with the manufacturer's instructions.

### Immunoprecipitation (IP) assay

Cells were harvested and lysed by using the IP lysis buffer (10006D, Invitrogen). After brief sonication, the whole cell lysates (WCL) were collected and centrifuged at 12000 × g for 10 min at 4 °C according to the manufacturer's instruction. 10% of the resulting protein was reserved as "input" and the remainder was immunoprecipitated. Then supernatant was incubated with anti-ZBP1 (1:200; AG-20B-0010; AdipoGen), anti-GSDMD (1:30; ab219800; Abcam), anti-GSDME (1:30; ab215191; Abcam) and anti-MLKL (1:50; 37705; Cell Signaling Technology) antibodies for 2 h, followed by the addition of Protein A/G Magnetic Beads (HY-K0202, MCE) and incubated overnight at 4 °C. The precipitates were washed three times with lysis buffer, boiled in sample buffer for 5 min and subjected to immunoblot assay with anti-NLRP6 antibody (1:1000; A15628; ABclonal) or anti-Ubiquitin antibody (1:1000; 10201; Proteintech). Odyssey Imager (LI-COR Biosciences) was used as a detection method according to the manufacturer's protocol.

### Immunohistochemistry (IHC)

4 μm paraffin-embedded sections of tumor tissue were subjected to dewaxing and rehydration. Antigen retrieval was achieved by heating the sections in a microwave oven using either citrate acid solution or EDTA. The sections were then treated with 3% hydrogen peroxide and blocked with goat serum. Primary antibodies were applied to the sections and incubated overnight at 4 °C. Primary antibodies against CD8 (1:400; 98941; Cell Signaling Technology), FOXP3 (1:400; 12653; Cell Signaling Technology), Granzyme B (1:200; 44153; Cell Signaling Technology) and Ki-67 (1:400; ab15580; Abcam) were used. Afterward, the sections were incubated with a secondary biotinylated IgG antibody followed by an anti-biotin-peroxidase reagent for a duration of 20 min. The detection of signals was accomplished using a DAB reagent (Mxb Biotechnologies), and the nuclei were counterstained with hematoxylin. Slide scanning was performed using a Pannoramic Midi scanner (3DHISTECH), and CaseViewer v. 2.4 (3DHISTECH) was used to create slide presentations. Histoscore analysis of the detection indicators was conducted utilizing QuantCenter software v. 12.3.2 (3DHISTECH)[57].

### Immunofluorescence staining

For immunofluorescence staining, the cells were fixed using 4% paraformaldehyde, permeabilized with 0.2% Triton X-100, and then blocked with 5% bovine serum albumin for 1 h. Following the blocking step, the cells were incubated overnight at 4 °C with the specific primary antibodies. After washing the cells three times with PBS, they were incubated with fluorophore-conjugated secondary antibodies (Abbkine) for 1 h. Nuclei were counterstained with DAPI (Beyotime). The acquired images were captured using a confocal laser scanning microscope and analyzed using FV10-ASW Viewer software (OLYMPUS). The specific primary antibodies used were anti-ZBP1 (1:400; AG-20B-0010; AdipoGen), anti-Z-DNA/Z-RNA (1:400; Ab00783-3.0; Absolute Antibody) and anti-HMGB1 (1:100; ab18256; Abcam).

### Multiplexed immunohistochemistry

Multiplexed immunohistochemistry was conducted utilizing the Opal 7-Color Manual IHC Kit (NEL811001KT; PerkinElmer, Hopkinton, MA, USA)[58]. The procedure included sequential steps such as dewaxing, alcohol hydration, AR6 buffer repair, PerkinElmer blocking buffer, incubation with primary antibodies, and tyrosine signal amplification (TSA) using the PerkinElmer Opal kit. These steps were repeated until the incubation with the final antibody, and the nuclei were stained with DAPI before sealing. Slides were scanned and typical images were

selected using the PerkinElmer Vectra software. The primary antibodies used in this study were CD8 (1:400; 98941; Cell Signaling Technology), PD-L1 (1:200; 64988; Cell Signaling Technology), F4/80 (1:200; 70076; Cell Signaling Technology) and FOXP3 (1:400; 12653; Cell Signaling Technology). The corresponding TSA dyes used were opal 620 (F4/80), opal 520 (CD8), opal 650 (PD-L1) and opal 570 (FOXP3).

## Stable cell line generation

Single guide RNAs (sgRNAs) targeting ZBP1, NLRP6, GSDMD, GSDME, and MLKL were constructed into the Lenti-CRISPRv2 vector. Lentivirus was generated in 293 T cells. 4T1 or 4MOSC1 cells were infected with lentivirus containing sgRNA[55]. After 36 h, 4T1 or 4MOSC1 cells were selected with 5 µg ml⁻¹ puromycin for 2 days. Cells were maintained in a culture medium containing puromycin (Sigma Aldrich). The sgRNA sequences are listed in Supplementary Data 1.

## Statistical analysis

The data were presented as mean ± standard error of the mean (s.e.m.). Multiple comparisons among more than two groups were performed using one-way ANOVA with Tukey's multiple comparisons test. Two-group comparisons were analyzed using a two-tailed Student's $t$ test. The survival benefit was determined using a log-rank (Mantel-Cox) test. All statistical analyses were conducted using GraphPad Prism software v.8.0 (GraphPad Software) and Excel 2016 software (Microsoft). A significance level of $P < 0.05$ was considered statistically significant for all types of analyses.

## Reporting summary

Further information on research design is available in the Nature Portfolio Reporting Summary linked to this article.

# Data availability

The data generated in this study are available within the Article, Supplementary Information, or Source Data file. Source data are provided with this paper.

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

## Acknowledgements

This work was financially supported by the National Natural Science Foundation of China 82273202 (Z.-J.S.), 82072996 (Z.-J.S.), the Fundamental Research Funds for the Central Universities (2042024kf0021, 2042022dx0003), National Key Research and Development Program 2022YFC2504200 (Z.-J.S.), and Interdisciplinary Innovative Foundation of Wuhan University XNJC202303. We acknowledge Shuyan Liang and Zhixin Qiu from Wuhan Biobank Co., Ltd for FACS assistance.

## Author contributions

S. Wang: Data curation, conceptualization, methodology, formal analysis, investigation, writing-original draft. A. Song: Data curation, conceptualization, methodology, formal analysis, writing-original draft. J. Xie: Data curation. Y.Y. Wang: Data curation, investigation. W.D. Wang: Data curation, investigation. M.J. Zhang: Data curation, methodology. Z.Z. Wu: Data curation, formal analysis. Q.C. Yang: Supervision. H. Li: Supervision. J.J. Zhang: Resources, conceptualization, project administration, funding acquisition. Z.J. Sun: Conceptualization, supervision, funding acquisition, writing-original draft, project administration, resources, writing-review and editing.

## Competing interests

The authors declare no competing interests.
