## [Peer Review File · Nature Communications]

REVIEWER COMMENTS

Reviewer #1 (Remarks to the Author):

This paper is very relevant to the field of OV therapy. The extensive data provide some new insights. The Figures are hard to read sometimes and the descriptions of the Results are often very brief.

Shuo Wang and colleagues present an in-depth study that combines oHSV and Fn-OMV to improve an θ tumor immunity. While oHSV has been used extensively to treat solid tumors based on tumor-targeted ly θ c replica θ on, it has become clear that induc θ on of an θ -tumor immunity is essen θ al for successful treatment. Treatment of uninfected tumors will rely on abscopal ac θ vi θ es. The induc θ on of immunity requires (i) widespread tumor killing with the release of tumor an θ gens and presenta θ on to and ac θ va θ on of immune cells, and (ii) countering an immunosuppressive tumor microenvironment. Many of these criteria were addressed in this paper. They confirmed that virus infec θ on and the ac θ va θ on of innate immune responses (e.g. IFN α) leads to high level produc θ on of IRG gene mRNA that can accumulate as endogenous ZRNA. Z-Nucleic acids ac θ vate ZBP-1 that in turn lead to a combina θ on of pyroptosis, apoptosis and necroptosis, i.e. PANoptosis. While ZBP-1 induc θ on by HSV is known, the authors show that PANoptosis enhances an θ -tumor immunity. ZBP-1 dependent PANoptosis depends on the release of the pyroptosis execu θ on protein GSDMD/E while necroptosis depends on the release of MLKL. Furthermore, they were able to enhance PANoptosis by combining oHSV with a bacterial membrane extract (Fn-OMV) that increased immunity and turned “cold” tumors into “hot” with cytokine/chemokine produc θ on and the release of ATP and LDH; both measures of cell membrane destruc θ on. Together these treatments led to strong an θ tumor responses including CD8 T cell recruitment, enhanced responsiveness to ICB and type 1 macrophage induc θ on. Overall, the authors were able to bring together known features of PANoptosis and oHSV infec θ on to effec θ ively treat a syngeneic model tumor in mice especially in the presence of CPB an θ bodies. Overall this is a substan θ al data package with many interes θ ng findings.

Comments:

1. While there are a lot or data presented, the key experiments regarding the induc θ on of

anti-tumor immunity were not formerly addressed. Namely, (i) demonstration that CD8 T cells were responsible for tumor rejection in Fig. 7, (ii) Memory CD8 T cells were induced as shown by tumor challenge experiments and (iii) that a second flank tumor was rejected following injection of the primary flank tumor (abscopal activity).

2. The figure legends are very brief and not always clear. For example, in Figure 1, I am not sure what the intersection plot is showing. What are numbers referring to? The human tumors seem to have been treated with oHSV but there is no description of this trial or issues. oHSV is not required in all cases to induce ZBP-1 making one wonder about how important this activity is in the clinical setting. Without clarification, this info could be dropped from the data set.

3. The paper is generally clearly written but the Results descriptions are parsimonious. While the conclusions seem to be supported by the data, experimental findings are often thinly described forcing the reader to study the figures to understand their significance. For example in Fig. 4, (J, K L), I am not sure what the MG132 data is telling us. Blocking proteasome activity with MG132 does not seem to affect Ubiquitination of MLKL. The author says that "Fn-OMV may regulate the ubiquitination process". Does this mean that no effect of MG132 indicates resistance to proteasome degradation and independent accumulation of MLKL. Also Fn-OMV activation is not required. What normally happens to MLKL? Is it degraded in proteasomes in the absence of Fn-OMV stimulation? Panel f shows an increase in Fn-OMV responsive genes but there are no statistical assessments of induction differences.

4. The production of ATP is not necessarily a good outcome since it can lead to immune tolerance. Local ectoenzymes CD39 and CD73 can increase production of adenosine (ADO) conferring immunosuppression and tumor aggression. Have they observed ADO activity.

5. The data sets are very small and difficult to see using the color coding. Red, dark red and orange are difficult to see. Presenting the data with better coding would be helpful. Many different abbreviations are used in the text that may be well known to many readers but the authors should be careful to define them at first use. One has to read all of the

methods to identify them all.

6. In Fig. 3, I would suggest eliminating tumor photos since measuring tumor volume is sufficient. This experiment is very short term. Do the tumors regrow?

7. The investigation nicely shows a reduction in FOXP3 suppressor T cells and increases in CD4 and CD8 T cells. ICB molecules CTLA-4, PD-1 and lag3 were all expressed and anti-PD1 greatly increased animal survival along with oHSV and Fn-OMV. The reduction in lung mets is difficult to interpret since this could result from destruction of the primary tumor by direct inoculation thus blocking metastasis. Did the lung tumors pre-exist before treatment.

8. The M2 to M1 conversion is a bit thin and there are other parameters for "hot" tumor designation, Nevertheless, recruitment of APCs and potentially immune memory T cells looks promising. Fig. 7 g is impressive.

9. The use of the Fn-OMV is a nice addition to the therapy. Unfortunately, however, it is not a defined substance so quality control for patient use may prove challenging. In the future the active ingredients will likely require definition. Does it just act as an adjuvant.

10. Have the authors tried dosing and timing of combination treatment?

11. A limitation of this approach is the need for direct intertumoral administration. Tumors may be difficult to access in some cases and widespread disease maybe resistant to this therapy.

12. Have the authors considered testing additional ICP blockades since other interactions like blocking CTLA-4 may be required.

Reviewer #2 (Remarks to the Author):

In this study, Wang et al., identified ZBP1 as a mediator of the oncolytic virus HSV-1 (oHSV-1)-induced PANoptotic tumor cell death and the subsequent release of immunogenic components in tumor cells using in vitro and in vivo models. The authors further investigated the synergistic effect of oHSV-1 and the fusobacterium outer membrane vesicle (Fn-OMV) and showed that this increased the expression of PANoptosis executioner proteins, including GSDMD, GSDME, and MLKL, and enhanced antitumor effects. Previous work has shown that Coley's toxin (mixed bacterial vaccine) can target the tumor

microenvironment and promote tumor regression in patients, and the present work is along similar lines. Therefore, the findings are of interest from a translational perspective; I have included a few comments that would further strengthen the study.

Major comments

1. HSV-1 is known to activate the AIM2-PANoptosome, with ZBP1 and Pyrin acting downstream (PMID: 34471287). However, the current study found downregulation of AIM2 (an IFN-inducible protein) during viral infection (Fig. 1a). The authors should explain this discrepancy and test the role of AIM2 in cell death with oHSV and the combination of oHSV and Fn-OMV to determine whether PANoptosis is solely reliant on ZBP1 in this system. This should be investigated in vitro and in vivo using AIM2 KO cells.

2. To determine whether Fn-OMV is unique in its ability to upregulate the expression of PANoptosis components, the author should also test TLR ligands such as the PAMPs LPS, Poly(I:C), and PAM3CSK4 or immune modulators (e.g., BCG) and adjuvants as controls in Fig. 4. The authors should also include immunoblots for ZBP1 and AIM2.

If these other PAMPs or immune modulators can induce similar upregulation of PANoptosis components, the advantage of utilizing Fn-OMV will need to be proven. The authors should also test these PAMPs/immune modulators in their in vitro (Fig. 5) and in vivo (Fig. 6–7) investigations on oHSV-induced cell death and tumor regression in comparison to Fn-OMV to identify the optimal combination. This would add significant value to the study.

Minor comments:

1. The authors should provide a specific definition for PANoptosis, as this is still a new term in the field and may not be well known to readers. PANoptosis is defined as a unique innate immune lytic, inflammatory programmed cell death pathway that is driven by caspases and receptor interacting protein kinases (RIPKs) and regulated by multiprotein PANoptosome complexes (PMID: 36782083, 36253067). Additionally, the initial description of ZBP1 as an upstream sensor that activates PANoptosis occurred in PMID: 27917412, and this reference should be cited.

2. The immunofluorescence for ZBP1 and Z-NA localization (Fig. 2) should be strengthened with additional controls. Nuclear localization of ZBP1 could be further confirmed by cellular fractionation to separate nuclear and cytosolic fractions followed by immunoblotting. Additionally, the Z-NA antibody specificity in recognizing Z-NAs and no other forms of nucleic acids should be confirmed. Details on the infection efficacy and viral titres in infected cells for Fig. 2f should also be provided.

3. The authors show that Fn and Fn-OMV increase the expression of PANoptosis executioners (Fig. 4); however, it is not clear whether the combination of oHSV with Fn or Fn-OMV results in a further increase in expression. This should be tested.

4. The authors should examine the effect of IFNAR1 neutralizing antibody treatment on cell death, ATP/LDH release, and NLRP6 protein expression in response to oHSV.

5. More methodological information is required for several of the analyses. In particular, the authors should explain how the protein expression profiles were obtained for each experiment (for example: Fig. 1a, c-e; Fig. 2b-d; Fig. 5g, i; and Fig. 6c, f), such as sequencing technique, viral dose, timepoints, etc. Additionally, the immunoblots in Fig. 4j-l should include a loading control.

6. The authors should take care when discussing "expression" vs "activation" in the text. "Expression" does not always indicate "activation." For example, line 144, "blotting analysis confirmed the activation of ZBP1 and NLRP6...", should be amended.

Reviewer #3 (Remarks to the Author):

This paper claimed that combination of oncolytic viral (OV) therapy with bacterial outer membrane vesicles (OMVs) would potentiate the therapeutic efficacy of current OV-based modalities, which have shown limitations in clinical trials. The key claims made in this paper are 1) the authors revealed the detailed mechanism by which oncolytic HSV activates ZBP-1 mediated cancer cell death and 2) combination of oHSV with Fn-OMV can further increase antitumor efficacy. These claims are mostly supported by sufficient data. And most experiments were well designed and their analysis or interpretation on the data are fair. However, I cannot recommend this paper in its current state for publication and would consider a revised draft. There are some critical issues to be considered and addressed before publication, which are noted below.

1. The findings that HSV can activate ZBP-1 and thus cause innate immune cytokine release, leading to infected cell death through inflammasome formation. In addition, the role of ZBP-1 pathway in cancer cell death is also known. In the context, it is natural to think that oHSV might induce cancer cell death through the ZBP-1-mediated pathway. What is a new finding or claim in this paper? Which can provide a new insight to the readers?

2. A key claim in this paper is that ISG mRNAs undergo conformational changes and are converted to corresponding Z-RNAs. However, no solid evidence for this claim was provided. According to a previous literature (Ref #21), it said that Z-type dsRNA structures could be enriched in the 3'-UTR of ISG mRNAs.

In this paper, ISG mRNAs were highly upregulated by oHSV treatment in cancer cells. It is also natural to think that the upregulated ISG mRNAs that contain Z-RNA structures in their 3'-UTRs can induce ZBP-1 activation, not through conversion of mRNAs into Z-RNAs.

3. In relation to the above comment 2, in Fig. 2a and 2f, the authors used anti-Z-DNA antibody and detected Z-DNA, NOT Z-RNA, in nucleus of the treated cancer cells. As noted above, the upregulation of ISG mRNAs itself can result in ZBP-1 activation. If so, why did the authors detect generation of Z-DNA in nucleus? What is direct correlation between the detection (presence) of Z-DNA in nucleus and ISG mRNAs? Then, what is a new finding in this paper?

4. Although Fn-OMVs showed meaningful synergistic effects with oHSV, the preparation of OMVs in general is highly costly and has limitation in mass production. In contrast, heat-treated dead bacteria itself is readily and easily prepared in large scale. In the context, the authors need to compare the therapeutic efficacy of oHSV + Fn-OMV with that of oHSV + heat-treated dead Fn. I would expect there is not much difference in the efficacy.

5. Z-NA seems not accurate expression. Instead, Z-DNA or Z-RNA depending on situation should be used at right place.

6. There are many abbreviations without spelling out or definition at their first appearance; for examples, PANaptosis, HNSCC, etc. A table or note for these abbreviations could be made separately.

Point-to-point responses to the comments

Reviewer #1

Shuo Wang and colleagues present an in-depth study that combines oHSV and Fn-OMV to improve antitumor immunity. While oHSV has been used extensively to treat solid tumors based on tumor-targeted lytic replication, it has become clear that induction of anti-tumor immunity is essential for successful treatment. Treatment of uninfected tumors will rely on abscopal activities. The induction of immunity requires (i) widespread tumor killing with the release of tumor antigens and presentation to and activation of immune cells, and (ii) countering an immunosuppressive tumor microenvironment. Many of these criteria were addressed in this paper. They confirmed that virus infection and the activation of innate immune responses (e.g., IFN α) leads to high level production of IRG gene mRNA that can accumulate as endogenous Z-RNA. Z-Nucleic acids activate ZBP-1 that in turn lead to a combination of pyroptosis, apoptosis and necroptosis, i.e. PANoptosis. While ZBP-1 induction by HSV is known, the authors show that PANoptosis enhances anti-tumor immunity. ZBP-1 dependent PANoptosis depends on the release of the pyroptosis execution protein GSDMD/E while necroptosis depends on the release of MLKL. Furthermore, they were able to enhance PANoptosis by combining oHSV with a bacterial membrane extract (Fn-OMV) that increased immunity and turned “cold” tumors into “hot” with cytokine/chemokine production and the release of ATP and LDH; both measures of cell membrane destruction. Together these treatments led to strong antitumor responses including CD8 T cell recruitment, enhanced responsiveness to ICB and type 1 macrophage induction. Overall, the authors were able to bring together known features of PANoptosis and oHSV infection to effectively treat a syngeneic model tumor in mice especially in the presence of CPB antibodies. Overall, this is a substantial data package with many interesting findings.

Response:

We are grateful to you for the positive and constructive comments. By responding to your comments in detail and revising the manuscript accordingly, we believe our manuscript has been significantly strengthened. We have conducted all additional experiments as suggested by you, and in accordance with your recommendations, we have supplemented our manuscript with detailed descriptions of the experimental results and explicit methodologies. All revisions are highlighted in red color in the revised manuscript and Supplementary Information. We extend our gratitude once again for the positive evaluation of our study and the numerous constructive suggestions that you have offered regarding our work.

Comment 1:

1. While there are a lot of data presented, the key experiments regarding the induction of anti-tumor immunity were not formerly addressed. Namely, (i) demonstration that CD8 T cells were responsible for tumor reject in Fig. 7, (ii) Memory CD8 T cells were induced as shown by tumor challenge experiments and (iii) that a second flank tumor was rejected following injection of the primary flank tumor (abscopal activity).

Response 1:

We are grateful to you for these insightful comments identifying limitations in the present study and offering constructive suggestions that will help strengthen it. In response to the three additional experiments recommended by you, we have rigorously conducted the corresponding

experiments according to your guidance.

According to your suggestions, we conducted *in vivo* CD8⁺ T cell depletion in mice using CD8 α monoclonal antibody (10 mg kg⁻¹; BE0004-1; Bio X Cell) to further validate the role of CD8⁺ T cells in tumor rejection observed in **Fig. 7**. The experimental results demonstrate that CD8⁺ T cells indeed mediate tumor rejection responses, as illustrated in **Fig. R1**.

Additionally, we observed elevated memory CD8⁺ T lymphocytes post-treatment. Following your suggestions, we performed repeat tumor challenge experiments in cured mice and naive mice to further demonstrate the long-term protective effects of treatment-induced memory T lymphocytes against tumors in mice. The experimental results suggest that treatment-induced memory T lymphocytes confer long-term protection against murine tumors, as depicted in **Fig. R2**.

Finally, upon injection of the primary flank tumor, we injected a second flank tumor as suggested by you to demonstrate abscopal activity. In the treatment group receiving oHSV+Fn-OMV+PD-L1, abscopal activity was observed with the rejection of a second flank tumor following injection of the primary flank tumor, as illustrated in **Fig. R3**.

Revision made:

In the revised manuscript, **Fig. R1** has been added as **Supplementary Fig. 15**.

Fig. R1 a Schematic representation of the immunization regimen for the killing assay and the timeline of animal experiments. **b** Average tumor growth curve depicted as mean \pm s.e.m. for each treatment group (n = 5 biological replicates). Statistical significance was evaluated using two-way ANOVA. Source data are provided as a Source Data file.

In the revised manuscript, **Fig. R2** has been added as **Supplementary Fig. 16**.

Fig. R2 a Establishment of the rechallenge model. Tumor volume curves (**b**) and

bioluminescence images (c) of naïve and oHSV+Fn-OMV+αPD-L1 treatment groups at different times; data are presented as mean ± s.e.m. (n = 5 biological replicates). Statistical significance was calculated via two-way ANOVA with Tukey’s multiple comparisons test. Source data are provided as a Source Data file.

In the revised manuscript, Fig. R3 has been added as Supplementary Fig. 17.

Fig. R3 a Schedule of oHSV+Fn-OMV+αPD-L1 combination therapy. The growth curves of primary (b) and distant tumor (c) of 4T1 tumor-bearing mice with different treatments. Data are presented as mean ± s.e.m. (n = 5 biological replicates). Statistical significance was calculated via two-way ANOVA with Tukey’s multiple comparisons test. Source data are provided as a Source Data file.

In the revised manuscript, the following statements were added in the Results section:

“We conducted *in vivo* CD8⁺ T cell depletion in mice using CD8α monoclonal antibody (10 mg kg⁻¹; BE0004-1; Bio X Cell) to further validate the role of CD8⁺ T cells in tumor rejection observed in Fig. 7f. The experimental results demonstrate that CD8⁺ T cells indeed mediate tumor rejection responses (Supplementary Fig. 15a, b). We performed repeat tumor challenge experiments in cured mice and naïve mice to further demonstrate the long-term protective effects of treatment-induced memory T lymphocytes against tumors in mice. The experimental results suggest that treatment-induced memory T lymphocytes confer long-term protection against murine tumors (Supplementary Fig. 16a-c). Finally, upon injection of the primary flank tumor, we injected a second flank tumor to demonstrate abscopal activity. In the treatment group receiving oHSV + Fn-OMV + PD-L1, abscopal activity was observed with the rejection of a second flank tumor following injection of the primary flank tumor (Supplementary Fig. 17a-c).”

Comment 2:

2. The figure legends are very brief and not always clear. For example, in Figure 1, I am not sure what the intersection plot is showing. What are numbers referring to? The human tumors seem to have been treated with oHSV but there is no description of this trial or tissues. oHSV

is not required in all cases to induce ZBP-1 making one wonder about how important this activity is in the clinical setting. Without clarification, this info could be dropped from the data set.

Response 2:

We appreciate you highlighting that our figure legends were overly concise with some aspects lacking comprehensive description. According to your suggestion, our team carefully re-examined the entire manuscript and provided more detailed context for the figure legends. We have aimed to ensure that each legend sufficiently explains the key experimental procedures, results being depicted, relevant statistical analyses, and other important methodological details to facilitate independent interpretation of the data.

We sincerely apologize for the lack of detailed description of Fig. 1b that led to confusion. The Venn diagram depicts which genes are upregulated across multiple cell types versus upregulated in only certain cell types upon addition of oHSV. Specifically, the two genes in the center indicate that two genes, ZBP1 and NLRP6, are commonly upregulated in all five cell lines infected with oHSV. Furthermore, the number "1" on the far left side signifies that one gene is upregulated only in the 4MOSC2 cell line but not in the other four cell types. We are sorry for the oversight that caused ambiguity. As suggested, we have provided a comprehensive explanation for Fig. 1b in the revised manuscript. Thank you for pointing out this limitation - it has helped improve the clarity of our work. We appreciate you taking the time to provide this feedback to strengthen our figure description.

We sincerely apologize for the lack of clarity regarding Fig. 1e that led to confusion. We only stimulated primary HNSCC tissues ex vivo with oHSV, rather than using tumor tissues from patients who received oHSV treatment. Considering this was an additional validation experiment, and to avoid ambiguity as suggested, we have removed this data from the revised manuscript. Furthermore, we would like to provide the following details about the experimental procedure:

Within 30 minutes of resection, freshly resected HNSCC tissues were cleaned and split into various sections (1-5 mm³) using a scalpel, followed by injection of 5% glucose control, and oHSV in a 5% glucose solution at multiple sites. Each segment was cultured in 24-well plates for 24 hours in 0.5 mL RPMI 1640 media.

Revision made:

In the revised manuscript, all figure legends have been expanded in detail. The following are the updated legends for Fig. 1, Fig. 2, and Fig. 3:

“Fig 1. oHSV induces ZBP1-mediated tumor cell death. Heatmap showing the expression of signature genes related to inflammasomes and certain key innate sensor-related genes in five types of tumor cells infected with oHSV (MOI = 1, 24 h) (a) and Venn diagram showed the increased gene intersections of five types of tumor cells infected with oHSV. Specifically, the two genes in the center indicate that two genes, Zbp1 and Nlrp6, are commonly upregulated in all five cell lines infected with oHSV (b). c Five types of tumor cells were infected with oHSV (MOI = 1, 24 h), cellular protein was analyzed by Western blotting to detect the expression level of ZBP1 and NLRP6. d oHSV-treated or untreated 4T1 tumor tissue lysates were analyzed of the expression of ZBP1 and NLRP6 by Western blotting. e Effect of ZBP1 and NLRP6 expression on relative cell viability in the 4T1 cells treated with oHSV (MOI = 1, 24 h). n = 3

biological replicates. n.s. not significant. f The effect of ZBP1 and NLRP6 expression on the injury of 4T1 cells after oHSV (MOI = 1, 24 h) treatment was detected by LDH release assay. n = 3 biological replicates. n.s. not significant. g The expression of Z-NA and ZBP1 in tumor cells after oHSV (MOI = 1, 24 h) treatment was evaluated using a Confocal Laser Scanning Microscope. Scale bars = 10 μ m. h, i Western blotting analysis assessing ZBP1 expression in the cytoplasm and nucleus of 4T1 cells with or without oHSV treatment (MOI = 1, 24 h); n = 3 biological replicates. j, k Western blotting analysis assessing ZBP1 expression in the cytoplasm and nucleus of 4MOSCI cells with or without oHSV treatment (MOI = 1, 24 h); n = 3 biological replicates. Statistical significance was determined using two-tailed Student's t-test in e, f, i, k. Data represent the mean \pm s.e.m. Scale bar, 10 μ m. Source data are provided in the Source Data file.”.

“**Fig 2. oHSV activates ZBP1 by inducing the accumulation of Z-RNA.** a Heatmap showing the expression of signature genes related to ISG mRNA in oHSV-treated (MOI = 1, 24 h) 4T1 (left) and 4MOSCI (right) cells. n = 3 biological replicates. b Heatmap indicated the expression of signature genes related to ISG mRNA with high editing indices in oHSV-treated (MOI = 1, 24 h) 4T1 (left) and 4MOSCI (right) cells. n = 3 biological replicates. c Heatmap demonstrated the influence of the expression of ISG mRNA with high editing indices on oHSV-treated (MOI = 1, 24 h) 4T1 cells after IFNAR-1 blockade. n = 3 biological replicates. d 4T1 cells were infected with oHSV (MOI = 1, 24 h) and IFNAR-1 blockade, cellular protein was analyzed by Western blotting to detect the expression level of ZBP1. e The expression of Z-NA and ZBP1 in 4T1 tumor cells after oHSV (MOI = 1, 24 h) alone and oHSV combined with DNase I (25 U mL⁻¹) treatment was evaluated using a Confocal Laser Scanning Microscope. Scale bars = 10 μ m. f The expression of Z-RNA in tumor cells after oHSV (MOI = 1, 24 h) treatment and IFNAR-1 blockade was evaluated using a Confocal Laser Scanning Microscope. Scale bars = 10 μ m. g Model diagram illustrating the upregulation of ZBP1 by oHSV through the induction of Z-RNA accumulation. The release of LDH (left) and ATP (right) from 4T1 (h) tumor cells after oHSV (MOI = 1, 24 h) treatment and IFNAR-1 blockade. n = 3 biological replicates. Statistical significance was determined using two-tailed Student's t-test in h. Data represent the mean \pm s.e.m. Scale bar, 10 μ m. Source data are provided in the Source Data file.”.

“**Fig 3. oHSV induces ZBP1-mediated PANoptosis in tumor cells.** 4T1 (a) and 4MOSCI (b) tumor cells were infected with oHSV (MOI = 1) for 24 h, cellular protein was analyzed by Western blotting to detect the effect of ZBP1 expression on PANoptosis-related executive proteins. Effect of ZBP1, GSDMD, GSDME and MLKL expression on relative cell viability in the 4T1 (c) and 4MOSCI (d) cells treated with oHSV (MOI = 1) for 24 h. n = 3 biological replicates. n.s. not significant. e Effect of ZBP1, GSDMD, GSDME and MLKL expression on ATP (left) and LDH (right) release in 4T1 tumor cells after oHSV (MOI = 1, 24 h) treatment. n = 3 biological replicates. f Schematic illustration of injection of 4T1 tumor cells and the effect of ZBP1, GSDMD, GSDME and MLKL expression on 4T1 tumor size after oHSV treatment. n = 5 biological replicates. g Schematic illustration of injection of 4T1 tumor cells and the effect of ZBP1, GSDMD, GSDME and MLKL expression on 4MOSCI tumor size after oHSV treatment. n = 5 biological replicates. Statistical significance was determined using two-tailed Student's t-test in c, d, e and two-way ANOVA with Tukey's multiple comparisons test in f and g. Data represent the mean \pm s.e.m. Source data are provided in the Source Data file.”.

Comment 3:

3. The paper is generally clearly written but the Results descriptions are parsimonious. While the conclusions seem to be supported by the data, experimental findings are often thinly described forcing the reader to study the figures to understand their significance. For example, in Fig. 4, (J, K L), I am not sure what the MG132 data is telling us. Blocking proteasome activity with Mg132 does not seem to affect Ubiquitination of MLKL. The author says that “Fn-OMV may regulate the ubiquitination process”. Does this mean that no effect of MG132 indicates resistance to proteasome degradation and independent accumulation of MLKL. Also Fn-OVM activation is not required. What normally happens to MLKL? Is it degraded in proteasomes in the absence of Fn-OMV stimulation? Panel f shows an increase Fn-OMV responsive genes but there are no statistical assessments of induction differences.

Response 3:

We are grateful to you for highlighting the need for more detailed reporting of our results. We sincerely apologize for any confusion caused by insufficient explanation. According to your constructive suggestion, we have thoroughly elaborated on the results section to ensure comprehensive description of all key experimental findings.

We sincerely apologize for the insufficiently detailed description of Figures 4, (j, k, l) and lack of rigor in the overall experimental design of Figure 4, (g, h, i, j, k, l), which led to confusion. Following your insightful suggestion, we have supplemented this section as follows:

First, we evaluated the impact of MG132 on the degradation process of GSDMD, GSDME, and MLKL proteins (results shown in **Fig. R4**). Our findings revealed that MLKL is subject to degradation in the absence of MG132, whereas MG132 administration leads to the suppression of MLKL degradation, signifying that MLKL follows a proteasome-mediated degradation pathway.

Furthermore, as suggested, we removed MG132 as a variable in IP experiments and independently assessed the impact of *Fn*-OMV on ubiquitination of GSDMD, GSDME and MLKL proteins (results shown in **Fig. R5**). The results demonstrated that *Fn*-OMV suppressed ubiquitination of these proteins.

In response to your suggestion, we have conducted quantitative analysis of **Fig. 4f** and included the results in the revised manuscript (results shown in **Fig. R6**). We greatly appreciate you for identifying this limitation and providing such constructive feedback.

Revision made:

In the revised manuscript, Fig. R4 has been added as Fig. 4

Fig. R4 Western blotting analysis was performed to evaluate the protein stability in 4T1 cells of GSDMD (**g**), GSDME (**h**), and MLKL (**i**) upon treatment with *Fn*-OMV ($1\mu\text{g ml}^{-1}$) and MG132 ($10\mu\text{M}$) in the presence of $50\mu\text{g ml}^{-1}$ cycloheximide (CHX).

In the revised manuscript, **Fig. R5** has been added as **Fig. 4**

Fig. R5 Immunoprecipitation (IP) and Western blotting analysis were performed to examine the ubiquitination and expression of GSDMD (**j**), GSDME (**k**), and MLKL (**l**) in Control and Gsdmd, Gsdme, or Mlkl-depleted 4T1 cells after 24 hours of *Fn*-OMV treatment.

In the revised manuscript, **Fig. R6** has been added as **Fig. S4**

Fig. R6 d Cellular protein was analyzed by Western blotting to detect corresponding quantitative analyses of GSDMD, GSDME, MLKL in 4T1 cells treated with *Fn* or *Fn*-OMV ($n = 3$ biological replicates). Statistical significance was calculated via one-way ANOVA with Tukey's multiple comparisons test. n.s. not significant. Source data are provided as a Source Data file.

In the revised manuscript, the following statements were added in the Results section:

“Heatmap showing the RNA expression level of signature genes related to inflammasomes and certain key innate sensor-related genes in five types of tumor cells infected with oHSV (MOI = 1, 24 h) (Fig. 1a). The Venn diagram depicts which genes are upregulated across multiple cell types versus upregulated in only certain cell types upon addition of oHSV (Fig. 1b). Specifically, the two genes in the center indicate that two genes, *Zbp1* and *Nlrp6*, are commonly upregulated in all five cell lines infected with oHSV (Fig. 1b). Furthermore, the number “1” on the far left side signifies that one gene is upregulated only in the 4MOSC2 cell line but not in the other four cell types (Fig. 1b). The results demonstrated that only the transcription levels of *Zbp1* and *Nlrp6* were upregulated in all five cell lines upon oHSV infection (Fig. 1a, b).”

“Notably, an excessive generation of ISG mRNAs, particularly those with a high editing index, can form Z-RNA within the 3' UTRs, thus activating ZBP1²¹. We examined the expression of ISG mRNAs in 4T1 and 4MOSC1 cells during oHSV infection and observed a significant upregulation of ISG mRNAs, along with an elevation of ISG mRNAs containing Z-prone regions within 3'-UTR (Fig. 2a, b). We initiated our protocol by incorporating DNase I to decompose DNA, followed by fluorescent detection, which revealed that DNase I treatment did not substantially alter the fluorescent signals of the Z-RNA. This indicates that the accumulation of

Z-NA is largely attributed to Z-RNA rather than Z-DNA (Fig. 2e). We observed that blockade of IFNAR-1 resulted in a reduced number of oHSV-induced tumor cell deaths (Supplementary Fig. 2b). Moreover, the blockade of IFNAR-1 led to decreased oHSV-induced release of LDH and ATP (Fig. 2h and Supplementary Fig. 2c)."

"Then, we detected and observed that Fn-OMV could also increase the expression level of PANoptosis execution proteins (Fig. 4f and Supplementary Fig. 4d), suggesting that Fn may enhance the expression level of PANoptosis execution proteins through Fn-OMV. We independently examined the effects of MG132 and Fn-OMV on the degradation of the GSDMD, GSDME, and MLKL proteins. Our results indicate that degradation of GSDMD, GSDME, and MLKL occurs in the absence of MG132 stimulation, while MG132 significantly inhibits their degradation (Fig. 4g-i), suggesting that these proteins are degraded via the proteasome pathway. Fn-OMV also significantly inhibits the degradation of the GSDMD, GSDME, and MLKL proteins (Fig. 4g-i). Further experiments revealed that Fn-OMV suppresses the ubiquitination of these proteins (Fig. 4j-l). Therefore, we propose that Fn-OMV enhances the expression levels of these proteins by affecting their ubiquitination."

"We also tested the impact of combined oHSV and Fn-OMV treatment on the expression of PANoptosis executioners (Supplementary Fig. 5). The results showed that compared to oHSV alone, combination of Fn-OMV with oHSV led to higher expression levels of PANoptosis executioners."

"Based on observed post-treatment increases in CD8⁺ T cell infiltration, the upregulation of inflammatory markers in the tumor microenvironment, and an enhanced response to ICB as shown in Fig. 7g, we hypothesize that treated tumors demonstrate features consistent with immunogenicity or "hot" tumors (Fig. 6k)."

"We conducted in vivo CD8⁺ T cell depletion in mice using CD8 α monoclonal antibody (10 mg kg⁻¹; BE0004-1; Bio X Cell) to further validate the role of CD8⁺ T cells in tumor rejection observed in Fig. 7f. The experimental results demonstrate that CD8⁺ T cells indeed mediate tumor rejection responses (Supplementary Fig. 15a, b). We performed repeat tumor challenge experiments in cured mice and naive mice to further demonstrate the long-term protective effects of treatment-induced memory T lymphocytes against tumors in mice. The experimental results suggest that treatment-induced memory T lymphocytes confer long-term protection against murine tumors (Supplementary Fig. 16a-c). Finally, upon injection of the primary flank tumor, we injected a second flank tumor to demonstrate abscopal activity. In the treatment group receiving oHSV + Fn-OMV + PD-L1, abscopal activity was observed with the rejection of a second flank tumor following injection of the primary flank tumor (Supplementary Fig. 17a-c). We conducted additional experiments combining our oHSV/Fn-OMV treatment with CTLA-4 blockade. The results demonstrated good synergistic effects when oHSV/Fn-OMV treatment was combined with CTLA-4 blockade (Supplementary Fig. 18a-c). However, the reduction in lung metastases could potentially result from destruction of the primary tumor by direct inoculation rather than a direct effect on metastasis. To rule out the influence of primary tumor burden, we designed additional lung metastasis experiments.

Specifically, we first injected non-luciferase 4T1 cells subcutaneously in the back, then performed intravenous injection of luciferase-expressing 4T1-luc cells one day after treatment completion. Bioluminescence imaging was used to assess the impact of treatment on lung metastasis. The results demonstrated treatment could indeed reduce lung metastasis

(Supplementary Fig. 19a, b).”

Comment 4:

4. The production of ATP is not necessarily a good outcome since it can lead to immune tolerance. Local ectoenzymes CD39 and CD73 can increase production of adenosine (ADO) conferring immunosuppression and tumor aggression. Have they observed ADO activity.

Response 4:

We greatly appreciate your constructive comment. As you rightly pointed out, increased ATP production may not always confer favorable antitumor immunity since local ectoenzymes can metabolize ATP to adenosine (ADO) to exert immunosuppression. We neglected to discuss this important aspect in our original manuscript. Following your valuable suggestion, we have added corresponding content for discussion in the revised manuscript.

Immune stimulatory processes are invariably accompanied by negative feedback mechanisms. For example, expression of negative immune checkpoint molecules like PD-L1 and CTLA4 were elevated with our oHSV and *Fn*-OMV treatment approach. Consistently, our sequencing data showed CD73 levels increased as anticipated post-treatment. As you foresaw, ATP would ultimately be converted to ADO through the actions of CD39 and CD73. While we detected increased ADO levels with rising ATP, the overall ATP/ADO ratio was significantly elevated upon treatment (results shown in **Fig. R7**). As one marker of immunogenic cell death, elevated ATP can effectively promote anti-tumor immunity.

Fig. R7 The quantitative examination of ATP (a) and ADO (b) secretion in the 4T1 cells supernatants after cells treated with oHSV + *Fn*-OMV treatments. c Overall ATP/ADO ratio in control and oHSV + *Fn*-OMV treatments group. n = 3 biological replicates. Data represent the mean \pm s.e.m.

The immunosuppressive effects of elevated ADO should not be overlooked. As we discussed in the revised manuscript, negative feedback regulation via mechanisms such as increased CD73 expression and adenosine production may dampen antitumor responses to some degree. Moving forward, it will be important to explore combination strategies incorporating ADO-generating enzyme inhibitors to potentially further enhance therapeutic outcomes. Investigating approaches to overcome ADO-mediated tolerance induction will likely strengthen our treatment regimen. We appreciate you prompting this forward-looking perspective, which will guide more impactful experimentation going forward.

Revision made:

In the revised manuscript, the following statements were added in the Discussion section:

“In this study, stimulation with oHSV led to an increase in ATP levels; however, it is noteworthy that the enhanced production of ATP may not always confer beneficial antitumor immunity, as ectoenzymes CD39 and CD73 can metabolize extracellular ATP into adenosine (ADO), exerting immunosuppressive effects. The process of immune stimulation is invariably accompanied by negative feedback mechanisms. For instance, treatment with our oHSV and Fn-OMV approach results in the upregulation of inhibitory immune checkpoint molecules such as PD-L1 and CTLA-4. Future research should further investigate the impact of ATP and ADO on overall therapeutic outcomes.”

Comment 5:

5. The data sets are very small and difficult to see using the color coding. Red, dark red and orange are difficult to see. Presenting the data with better coding would be helpful. Many different abbreviations are used in the text that maybe be well known to many readers but the authors should be careful to define them at first use. One has to read all of the methods to identify them all.

Response 5:

We greatly appreciate your constructive comment. Following your valuable suggestion, we have revised the figures to employ different shapes to distinguish groups, aiming to enhance visual interpretation.

We sincerely apologize for the oversight of not defining abbreviations upon their first use, which likely caused inconvenience for readers. To address this, we have now included a note listing all abbreviations along with their full terms and definitions. This note is positioned at the end of the manuscript. We have also ensured that each abbreviation is spelled out and defined in the text at its first use.

Revision made:

In the revised manuscript, Fig. R8 has been added as Fig. 1.

Fig.R8 e Effect of ZBP1 and NLRP6 expression on relative cell viability in the 4T1 cells treated with oHSV (MOI = 1, 24 h). n = 3 biological replicates. n.s. not significant. **f** The effect of ZBP1 and NLRP6 expression on the injury of 4T1 cells after oHSV (MOI = 1, 24 h) treatment was detected by LDH release assay. n = 3 biological replicates. n.s. not significant.

In the revised manuscript, **Fig. R9** has been added as **Fig. 3**.

Fig.R9 e Effect of ZBP1, GSDMD, GSDME and MLKL expression on ATP (left) and LDH (right) release in 4T1 tumor cells after oHSV treatment (n = 3 biological replicates).

In the revised manuscript, **Fig. R10** has been added as **Fig. 4**

Fig.R10 a 4T1 cells were infected with different bacterial titers (MOI = 0, 1, 10, 20, 50, 100) for 24 h, cellular protein was analyzed by Western blotting to detect the corresponding quantitative analyses of GSDMD, GSDME and MLKL (n = 3 biological replicates).

In the revised manuscript, **Fig. R11** has been added as **Fig. 6**

Fig.R11 f Immunological profile of tumour-draining lymph nodes. This includes the frequency of the lymph nodes CD80⁺ CD86⁺ population in CD11c⁺ cells in 4T1 (left) and 4MOSC1 (right) tumors (n = 5 biological replicates).

We have added the abbreviation note in the manuscript:

“*PANoptosis*: pyroptosis, apoptosis, necroptosis; *OVs*: oncolytic viruses; *HSV-1*: herpes simplex virus-1; *oHSV*: HSV-1-based *OVs* (*oHSV*); *OMV*: outer membrane vesicles; *Fn*: *Fusobacterium nucleatum*; *ICB*: immune checkpoint blockade; *RIPKs*: receptor interacting

protein kinases; ZBP1: Z-DNA binding protein 1; Z-NA: Z-type nucleic acids; MLKL: mixed lineage kinase domain-like protein; GSDMD/E: gasdermin D/E; AIM2: absent in melanoma 2; ISG: interferon-stimulated gene; qRT-PCR: quantitative real-time polymerase chain reaction; ADARI: adenosine deaminase RNA specific 1; EREs: endogenous retroviral elements; OSCC: oral squamous cell carcinoma; TDLNs: tumor draining lymph nodes; ELISA: enzyme-linked immunosorbent assay; TEM: transmission electron microscope; DLS: dynamic light scattering; IHC: immunohistochemistry; Tregs: regulatory T cells; TILs: tumor-infiltrating lymphocytes; TNF- α : tumor necrosis factor alpha; HMGB1: high mobility group box 1.”

Comment 6:

6. In Fig. 3, I would suggest eliminating tumor photos since measuring tumor volume is sufficient. This experiment is very short term. Do the tumors regrow?

Response 6:

We greatly appreciate your suggestion. In regard to your question about tumor regrowth after oHSV administration, our prolonged observation suggests that recurrence occurs in tumors treated solely with oHSV (results shown in **Fig. R12**). Therefore, investigating combinatory therapeutic approaches is necessary, and in this study, the combination of *Fn*-OMV with oHSV significantly inhibited tumor growth and markedly extended the survival time in mice.

According to your suggestion, we have removed the tumor photos from Fig. 3 in the revised manuscript (results shown in **Fig. R13**).

Fig.R12 a Experimental schedule for 4T1 tumor inoculation and treatment with oHSV. **b** Tumor volume of 4T1 tumor-bearing mice with oHSV (n = 5 biological replicates).

Revision made: In the revised manuscript, **Fig. R13** has been added as **Fig. 3**

Fig. R13 f Schematic illustration of injection of 4T1 tumor cells and the effect of ZBP1, GSDMD, GSDME and MLKL expression on 4T1 tumor size after oHSV treatment (n = 5 biological replicates). **g** Schematic illustration of injection of 4MOSC1 tumor cells and the effect of ZBP1, GSDMD, GSDME and MLKL expression on 4MOSC1 tumor size after oHSV treatment (n = 5 biological replicates).

Comment 7:

7. The investigation nicely shows a reduction in FOXP3 suppressor T cells and increases in CD4 and CD8 T cells. ICB molecules CTLA-4, PD-1 and lag3 were all expressed and anti-PD-1 greatly increased animal survival along with oHSV and Fn-OMV. The reduction in lung mets is difficult to interpret since this could result from destruction of the primary tumor by direct inoculation thus blocking metastasis. Did the lung tumors pre-exist before treatment.

Response 7:

We greatly appreciate your constructive comment. As you suggested, the reduction in lung metastases could potentially result from destruction of the primary tumor by direct inoculation rather than a direct effect on metastasis. To rule out the influence of primary tumor burden, we designed additional lung metastasis experiments following your suggestion.

Specifically, we first injected non-luciferase 4T1 cells subcutaneously in the back, then performed intravenous injection of luciferase-expressing 4T1-luc cells one day after treatment completion. Bioluminescence imaging was used to assess the impact of treatment on lung metastasis. The results demonstrated treatment could indeed reduce lung metastasis, as shown in **Fig. R14**.

Furthermore, to observe if lung tumors pre-existed before treatment, we subcutaneously injected luciferase-expressing 4T1-luc cells and monitored the lungs on Day 7 and 14. No lung tumors were observed on Day 7 and 14, indicating lung metastases were not present before treatment initiation, as illustrated in **Fig. R15**.

Fig. R15 a Schematic showing the experiment treat mice with the 4T1-Luc experimental model of metastasis at an early stage. BALB/c mice (n = 3 biological replicates) were implanted subcutaneously with 1×10^6 4T1 mammary carcinoma cells on right hind flanks to discover if the tumor in the back has metastasized to the lungs. **b** Bioluminescence images tracking the spreading and growth of intravenously injected 4T1-Luc cells in BALB/c mice (n = 3 biological replicates) treated with different treatment strategies on days 7 and 14.

Revision made:

In the revised manuscript, **Fig. R14** has been added as **Supplementary Fig. 19**

Fig. R14 a Schematic showing the experiment treat mice with the 4T1 experimental model of metastasis at an early stage. BALB/c mice ($n = 5$ biological replicates) were implanted subcutaneously with 1×10^6 4T1 mammary carcinoma cells on right hind flanks and after 13 days consequently implanted intravenously with 5×10^5 4T1 cells to evaluated the effect of different treatment strategies on tumor metastasis in a lung metastasis mouse model. When primary tumor volumes were 100mm^3 , mice received different treatments. **b** Bioluminescence images tracking the spreading and growth of intravenously injected 4T1-Luc cells in BALB/c mice ($n = 5$ biological replicates) treated with different treatment strategies on days 30.

In the revised manuscript, the following statements were added in the Results section:

“However, the reduction in lung metastases could potentially result from destruction of the primary tumor by direct inoculation rather than a direct effect on metastasis. To rule out the influence of primary tumor burden, we designed additional lung metastasis experiments. Specifically, we first injected non-luciferase 4T1 cells subcutaneously in the back, then performed intravenous injection of luciferase-expressing 4T1-luc cells one day after treatment completion. Bioluminescence imaging was used to assess the impact of treatment on lung metastasis. The results demonstrated treatment could indeed reduce lung metastasis (Supplementary Fig. 19)”

Comment 8:

8. The M2 to M1 conversion is a bit thin and there are other parameters for “hot” tumor designation, Nevertheless, recruitment of APCs and potentially immune memory T cells looks promising. Fig. 7 g is impressive.

Response 8:

We greatly appreciate your constructive comment. We wholeheartedly agree with your point that the M2 to M1 phenotypic conversion is an important but not exclusive aspect of the transition of the tumor microenvironment towards a more immunogenic state. Following your suggestion, we recognize that a more comprehensive characterization of the tumor microenvironment is necessary to fully substantiate the classification of tumors as “hot.” In addition to the M2 to M1 shift, we have measured several other parameters including the infiltration of effector T cells, the expression of pro-inflammatory cytokines, the activation of dendritic cells, and changes in memory T cells (with experimental results shown in **Fig. R16**).

The data indicate increased infiltration of effector T cells and a significant rise in the expression of pro-inflammatory cytokines in tumors following combination therapy. Furthermore, there is a notable increase in the number of activated dendritic cells as well as memory T cells in the combination therapy group.

Thank you again for your valuable comments. We are committed to improving our manuscript and addressing the concerns raised.

Fig. R16 **a** Heatmap shows a 17-gene expression profile associated with inflammation produced in mice treated with control, oHSV alone, *Fn*-OMV alone, or a combination of oHSV and *Fn*-OMV (n = 3 biological replicates). **b** CD8⁺ T cell populations in the tumors were measured using flow cytometry, and quantitative analyses were performed in 4T1 (left) and 4MOSC1 (right) tumors (n = 5 biological replicates). **c** Heatmap shows a selected four-gene expression signature represented by genes highly associated with CD8⁺ T cell activation (n = 3 biological replicates) treated with control, oHSV alone, *Fn*-OMV alone, or in combination. **d** Representative images of immunohistochemical analysis of CD8 and Granzyme B (GZMB) expression in different treatment groups of 4T1 tumors (left) and 4MOSC1 tumors (right), Scale bars, 50 μ m. **e** The expression of CD8 (green), PD-L1 (red), F4/80 (white), and Foxp3 (orange)

was observed in 4T1 tumors using multiplex immunohistochemistry (mIHC), Scale bars, 50 μm . **f** Immunological profile of tumour-draining lymph nodes. This includes the frequency of the lymph nodes $\text{CD80}^+ \text{CD86}^+$ population in CD11c^+ cells in 4T1 (left) and 4MOSC1 (right) tumors ($n = 5$ biological replicates). **g** CD44^+ and CD62L^+ T cell populations in the tumour-draining lymph nodes were measured using flow cytometry, and quantitative analyses were performed ($n = 5$ biological replicates).

Revision made:

In the revised manuscript, the following statements were added into the Results section:

“Based on observed post-treatment increases in CD8^+ T cell infiltration, the upregulation of inflammatory markers in the tumor microenvironment, and an enhanced response to immune checkpoint blockade as shown in Fig. 7g, we hypothesize that treated tumors demonstrate features consistent with immunogenicity or “hot” tumors (Fig. 6k).”

Comment 9:

9. The use of the Fn-OMV is a nice addition to the therapy. Unfortunately, however, it is not a defined substance so quality control for patient use may prove challenging. In the future the active ingredients will likely require definition. Does it just act as an adjuvant.

Response 9:

We sincerely appreciate your constructive comment, which will strengthen our work. We fully agree with the point raised regarding the challenges of quality control for patient applications due to Fn-OMV not being a well-defined entity, as you insightfully noted. Following your valuable suggestion, in the revised manuscript we have included a discussion of this limitation of our current study. Additionally, following your suggestion, in future work we aim to explore identifying the active components responsible for the observed effects.

Fn-OMV could serve as an ideal adjuvant due to its carriage of various bacterial components such as LPS, proteins and nucleic acids, which can activate multiple cellular receptors to promote activation of innate and adaptive immunity. Furthermore, the results in **Fig. R17** demonstrate that Fn-OMV alone also exhibited antitumor effects, with a significant reduction in tumor volume compared to the Control group. The ability of Fn-OMV to directly inhibit cancer cell growth as well as stimulate antitumor immunity likely synergistically contributed to the enhanced therapeutic efficacy observed upon combination with oHSV.

Fig. R17 a Schematic representation of 4T1 (upper) and 4MOSC1 (lower) tumor inoculation and treatment with oHSV and Fn-OMV for the killing assay. **b** Average 4T1 (left) and 4MOSC1 (right) tumor growth curve depicted as mean \pm s.e.m. for each treatment group ($n = 5$ biological replicates). Statistical significance was evaluated using two-way ANOVA.

Revision made:

In the revised manuscript, the following statements in Discussion were added:

“The use of the Fn-OMV is a nice addition to the therapy. However, it is not a defined substance so quality control for patient use may prove challenging. In the future the active ingredients will likely require definition.”

Comment 10:

10. Have the authors tried dosing and timing of combination treatment?

Response 10:

Thanks for your insightful comment. In our pre-experiment, we evaluated the dosages and regimens of the combination therapy. The specific experimental schemes and results are shown in the **Fig. R18**.

The analysis of four distinct timing protocols revealed that the regimen involving administration of *Fn*-OMV on days 7, 9, and 11, complemented by oHSV on days 8, 10, and 12, significantly outperformed the other schedules. Notably, this regimen led to the maximum reduction in tumor volume and the highest increase in CD8⁺ T cell infiltration within the tumor microenvironment, markers indicative of a robust antitumor response. These encouraging results underpinned our decision to adopt this timing scheme for subsequent experimental phases.

In parallel, we tested three separate *Fn*-OMV dosages to elucidate their impact on the antitumor activity. It was observed that dosages above 5 μg of *Fn*-OMV did not result in further significant improvements in tumor size reduction or the recruitment of immune cells to the tumor site, suggesting a plateau in the therapeutic effect beyond this concentration. Hence, guided by these comprehensive data, we delineated the dosing strategy and quantities detailed in the current manuscript.

Building upon the findings presented, future experiments will be essential to refine and further optimize the dosages and schedules of our combination therapy. While we have established a promising therapeutic protocol, the complex dynamics of tumor biology and host immune responses necessitate a continuous search for even more efficacious and tailored treatment strategies.

Subsequent studies will aim to explore a wider range of dosages, potentially including sub-therapeutic and supra-therapeutic concentrations, to delineate a dose-response curve with greater precision. Investigations could also extend to different tumor models and host systems to validate the generalizability and enhance the translatability of our findings.

Fig. R18 a Schematic representation of 4T1 tumor inoculation and treatment with oHSV and *Fn*-OMV for the killing assay under different dosing sequence strategies. **b** Average 4T1 tumor growth curve depicted as mean \pm s.e.m. for each treatment group ($n = 5$ biological replicates). Statistical significance was evaluated using two-way ANOVA. **c** Representative images of immunohistochemical staining of CD8 expression in different treatment groups of 4T1 tumors, Scale bars, 50 μ m. **d** Schematic representation of 4T1 tumor inoculation and treatment with oHSV and *Fn*-OMV for the killing assay under different dosages. **e** Average 4T1 tumor growth curve depicted as mean \pm s.e.m. for each treatment group ($n = 5$ biological replicates). Statistical significance was evaluated using two-way ANOVA. **f** Representative images of immunohistochemical staining of CD8 expression in different treatment groups of 4T1 tumors, Scale bars, 50 μ m. **g** Body weight curve of different groups.

Comment 11:

11. A limitation of this approach is the need for direct intertumoral administration. Tumors may be difficult to access in some cases and widespread disease maybe resistant to this therapy.

Response 11:

We sincerely appreciate you for pointing out the limitations in our work. Based on your valuable suggestion, we have added a discussion on the limitations of the delivery approach in

the revised manuscript.

We fully agree with your perspective that while direct intratumoral injection demonstrated proof-of-concept in our animal studies, this route of administration faces translational challenges. For example:

1. Access to primary or metastatic tumors may be restricted anatomically for some cancer types.

2. Multiple disseminated tumors in advanced patients would require enormous effort to target individually.

3. Intravenous delivery is safer and more practical but requires ensuring effective drug concentrations reach tumor sites.

To address these limitations, future work will need to explore alternative delivery methods capable of achieving therapeutic concentrations of our oHSV/*Fn*-OMV combination systemically. Possible strategies could involve engineered carriers, conjugation to tumor-targeting moieties, or adaptation for intravenous administration routes. Thank you for prompting consideration of these important translational hurdles - overcoming barriers to broader delivery will be critical for realizing clinical impact.

Revision made:

In the revised manuscript, the following statements in Discussion were added:

“A potential limitation of the current approach is the need for direct intratumoral injection to achieve localized therapeutic effects. For tumors that are difficult to access surgically or tumors with widespread metastatic disease, this requirement for direct administration may limit applicability. Further development is needed to optimize delivery methods that could enable broader application to additional cancer types and disease stages.”.

Comment 12:

12. Have the authors considered testing additional ICP blockades since other interactions like blocking CTLA-4 may be required.

Response 12:

We sincerely appreciate your valuable suggestion. Based on the synergistic effects observed when combining our oHSV/*Fn*-OMV therapy with PD-L1 blockade, as you insightfully suggested, blocking additional immune checkpoint molecules may produce similar synergy.

According to your suggestion, we conducted additional experiments combining our oHSV/*Fn*-OMV treatment with CTLA-4 blockade as shown in **Fig. R19**. The results demonstrated good synergistic effects when oHSV/*Fn*-OMV treatment was combined with CTLA-4 blockade.

Given the diverse resistance mechanisms that tumors can employ, it is essential to explore the effectiveness of multiple immune checkpoint inhibitors. We appreciate you for prompting us to consider this important avenue. If you have any further suggestions for additional testing that may strengthen our findings, please let us know.

Revision made:

In the revised manuscript, **Fig. R19** has been added as **Supplementary Fig. 18**

Fig. R19 Combination of *Fn*-OMV and oHSV enhances the effectiveness of anti-CTLA-4 immunotherapy. **a** Schedule of oHSV+*Fn*-OMV+αCTLA-4 combination therapy. **b** Tumor volume of 4T1 tumor-bearing mice with different treatments (n = 5 biological replicates). Statistical significance was calculated via two-way ANOVA with Tukey’s multiple comparisons test. The growth curves of percent survival (**c**) of 4T1 tumor-bearing mice with different treatments. Data are presented as mean ± s.e.m. (n = 5 biological replicates). Statistical significance was calculated via two-way ANOVA with Tukey’s multiple comparisons test. Source data are provided as a Source Data file.

In the revised manuscript, the following statements were added in the Results section:

“We conducted additional experiments combining our oHSV/*Fn*-OMV treatment with CTLA-4 blockade (10 mg kg⁻¹; BE0164; Bio X Cell). The results demonstrated good synergistic effects when oHSV/*Fn*-OMV treatment was combined with CTLA-4 blockade (Supplementary Fig. 18a-c).”

Reviewer #2

In this study, Wang et al., identified ZBP1 as a mediator of the oncolytic virus HSV-1 (oHSV-1)-induced PANoptotic tumor cell death and the subsequent release of immunogenic components in tumor cells using in vitro and in vivo models. The authors further investigated the synergistic effect of oHSV-1 and the fusobacterium outer membrane vesicle (Fn-OMV) and showed that this increased the expression of PANoptosis executioner proteins, including GSDMD, GSDME, and MLKL, and enhanced antitumor effects. Previous work has shown that Coley's toxin (mixed bacterial vaccine) can target the tumor microenvironment and promote tumor regression in patients, and the present work is along similar lines. Therefore, the findings are of interest from a translational perspective; I have included a few comments that would further strengthen the study.

Response:

We express our sincere gratitude to you for the positive and constructive comments. We also highly appreciate your suggestions for strengthening our work. In meticulously addressing each of your comments and making the appropriate revisions, we believe our manuscript has been significantly strengthened. We have conducted all additional experiments as suggested by you. In accordance with your suggestion, we have added the relevant experimental methodological information. All amendments have been clearly marked in red within the revised manuscript and the Supplementary Information to facilitate review. We deeply appreciate the opportunity to refine our study and are once again thankful for your affirmative assessment and the array of insightful recommendations you have provided for our research.

Major Comment 1:

1. HSV-1 is known to activate the AIM2-PANoptosome, with ZBP1 and Pyrin acting downstream (PMID: 34471287). However, the current study found downregulation of AIM2 (an IFN-inducible protein) during viral infection (Fig. 1a). The authors should explain this discrepancy and test the role of AIM2 in cell death with oHSV and the combination of oHSV and Fn-OMV to determine whether PANoptosis is solely reliant on ZBP1 in this system. This should be investigated in vitro and in vivo using AIM2 KO cells.

Response 1:

We sincerely appreciate your constructive comments. We apologize for not fully discussing AIM2's role in oHSV infection, which likely caused confusion for you and readers. According to your valuable suggestion, we have supplemented relevant discussion content and conducted relevant experiments to further explore AIM2's role in oHSV infection.

At the mRNA level, our original data showed that upon oHSV infection, AIM2 was upregulated in some tumor cell lines (4MOSC1, 4MOSC2) but downregulated in others (SCC7, 4T1, CT26), as shown in **Fig. R20**. Given that ZBP1 expression was upregulated at both the RNA and protein levels across all five tumor cell lines tested upon oHSV infection, we focused subsequent studies on further exploring ZBP1's role during oHSV infection. However, we did not neglect AIM2's potential role in oHSV infection throughout our study. We apologize again for the lack of relevant discussion in our initial manuscript.

Fig. R20 a Heatmap showing the expression of signature genes related to inflammasomes and certain key innate sensor-related genes in five types of tumor cells infected with oHSV. **b** Five types of tumor cells were infected with oHSV (MOI = 1) for 24 h, cellular protein was analyzed by Western blotting to detect the expression level of ZBP1.

Based on your suggestion, to further validate changes in AIM2 expression levels during oHSV infection, we performed Western blot analysis using an AIM2 antibody (63660, Cell Signaling Technology), as shown in **Fig. R21**. The results demonstrated that the changes in AIM2 protein expression were consistent with those previously observed at the mRNA level, namely that during oHSV infection AIM2 expression was upregulated in some tumor cell lines (4MOSC1, 4MOSC2) but downregulated in others (SCC7, 4T1, CT26).

Fig. R21 Five types of tumor cells were infected with oHSV (MOI = 1) for 24h, cellular protein was analyzed by Western blotting to detect the expression level of AIM2.

As you pointed out, "HSV-1 activates the AIM2 inflammasome, with ZBP1 and Pyrin acting downstream (PMID:34471287)". We also recognized this discrepancy, as theoretically AIM2 can sense invading pathogen (bacterial, viral) dsDNA and trigger upregulation. In that report (PMID:34471287), macrophages upregulated AIM2 at the protein level upstream of ZBP1 and pyrin after HSV1 or *F. novicida* infection. However, our current results in various cancer cell lines differ somewhat from this prior study.

Based on current findings, we speculated several possibilities for the discrepancies:

1. Role of AIM2 regulatory proteins: The organism has developed numerous regulatory proteins to ensure the homeostasis of AIM2. IFN α directly augments the expression of AIM2, yet conversely, upregulated AIM2 can attenuate IFN α production. Additionally, negative regulatory mechanisms, such as the action of TRIM11 which, upon DNA virus infection, associates with AIM2 to facilitate its degradation through selective autophagy (PMID: 27498865), are also in place. Such regulatory proteins might contribute to the downregulation of IFN α -induced AIM2 in certain neoplastic cells.
2. Viral antagonistic mechanisms: Viruses have developed strategies to evade host immune surveillance, which includes the epigenetic modulation of AIM2 expression. For instance, hepatitis B virus (HBV) has been shown to suppress the transcription of AIM2 in an effort to circumvent immune detection of HBV surface antigens (PMID: 27226092). Although AIM2 protein levels initially rise following human cytomegalovirus (HCMV) infection, they diminish after 24 hours, potentially as a result of viral immunoevasive actions (PMID: 28219398). HSV-1 VP22 impedes AIM2-dependent inflammasome assembly (PMID: 29447697), and concurrently, TRIM11 is implicated in targeting AIM2 for degradation (PMID: 27498865). Consequently, viral antagonistic mechanisms might also play a role in the downregulation of AIM2 within certain cancer cell contexts.
3. Differences in Cell Types and Immune Contexts: The expression of AIM2 shows considerable variability among different cell types and can be mutated or inactivated in various cancer cell lines, such as a high mutation rate of AIM2 observed in colorectal cancer (PMID: 24729378). During pathogenic invasion, AIM2 in macrophages and dendritic cells (DCs) can sense pathogen nucleic acids, leading to upregulation and activation that promote immune responses. However, in the tumor microenvironment, the upregulation of AIM2 is more often associated with immune suppression (PMID: 32053518). Moreover, the role of AIM2 is complex and unclear; some reports have described it as a tumor suppressor while others have linked it to the progression of various tumors (PMID: 34489308). Therefore, we posit that the interaction of various factors within the complex tumor microenvironment may lead to the downregulation of AIM2 expression in certain cancer cells during oHSV infection, contributing to the observed discrepancies from studies in macrophages.

Based on current results, we cautiously consider that the above factors may play a role. In accordance with your suggestion, we have added the corresponding content to the Discussion section of the manuscript. Going forward, we will further investigate the specific underlying mechanisms.

In addition, following your advice, we constructed AIM2 KO cells and examined AIM2's role in oHSV infection. The results demonstrated that AIM2 knockout had some impact on oHSV, as shown in **Fig. R22**. We sincerely apologize again for the lack of relevant discussion previously.

Fig. R22 a Schematic representation of 4MOSC1 tumor inoculation and treatment with oHSV for the killing assay. **b** The tumor growth curve showed the influence of Zbp1 and Aim2 expression on the oHSV therapeutic effect (n = 5 biological replicates). Statistical significance was evaluated using two-way ANOVA.

Revision made:

In the revised manuscript, the following statements were added in the Discussion section:

“It is known that HSV-1 activates the absent in melanoma 2 (AIM2) inflammasome in bone marrow-derived macrophages (BMDMs), with ZBP1 and Pyrin acting downstream¹⁹. Our findings suggest that during HSV-1 infection, AIM2 expression was upregulated in some tumor cell lines (4MOSC1, 4MOSC2) but downregulated in others (SCC7, 4T1, CT26), while ZBP1 expression was upregulated across all five tumor cell lines tested. Further investigation revealed that ZBP1-mediated PANoptosis plays an important role in the antitumor effects of oHSV in cancer. These findings indicate that innate sensors may be modulated differently during oHSV infection in tumor cells, as compared to their regulation in macrophages previously reported. We speculated several possibilities for the discrepancies: differential expression of regulatory proteins that balance AIM2 activity and maintain homeostasis⁴⁸; viral evasion tactics that suppress AIM2 to avoid immune detection⁴⁹; and the heterogeneous cellular and immune contexts within the tumor microenvironment that influence AIM2's roles⁵⁰. These dynamics could attribute to the observed downregulation of AIM2 in certain cancer scenarios. Previous studies have found that virus intracellular replication such as influenza virus (IAV) and vaccinia virus (VACV) can generate a substantial amount of Z-RNA to activate ZBP137,⁵¹ but it also been observed that intracellular replication of viruses like rhabdovirus (VSV) or poliovirus (EMCV) does not produce detectable Z-RNA³⁷. In this study, we have identified that oHSV primarily activates ZBP1 in tumor cells through the production of endogenous Z-RNA.”

Major Comment 2:

2. To determine whether Fn-OMV is unique in its ability to upregulate the expression of PANoptosis components, the author should also test TLR ligands such as the PAMPs LPS, Poly(I:C), and PAM3CSK4 or immune modulators (e.g., BCG) and adjuvants as controls in Fig. 4. The authors should also include immunoblots for ZBP1 and AIM2.

If these other PAMPs or immune modulators can induce similar upregulation of PANoptosis components, the advantage of utilizing Fn-OMV will need to be proven. The authors should also test these PAMPs/immune modulators in their in vitro (Fig. 5) and in vivo (Fig. 6–7) investigations on oHSV-induced cell death and tumor regression in comparison to Fn-OMV to identify the optimal combination. This would add significant value to the study.

Response 2:

We greatly appreciate your constructive comment. According to your valuable suggestion, we have conducted additional experiments:

1. To determine whether *Fn*-OMV is unique in its ability to upregulate the expression of PANoptosis components, we incorporated a spectrum of immune modulators, such as lipopolysaccharide (LPS), interleukin-2 (IL-2), interferon-gamma (IFN-gamma), polyinosinic:polycytidylic acid (Poly(I:C)), and PAM3CSK4, as comparative controls to evaluate their ability to alter PANoptosis-related protein levels. Our findings, as illustrated in **Fig. R23**, provide evidence that *Fn*-OMV, embodying a composite vehicle for an array of bacterial constituents including but not limited to proteins and nucleic acids, exhibits a superior capacity to upregulate the expression of proteins implicated in PANoptosis. This observation is distinctly contrasted with the effects elicited by the single adjuvants tested—LPS, IL-2, IFN-gamma, Poly(I:C), and PAM3CSK4—underscoring the potential of *Fn*-OMV as a more potent modulator of cellular pathways leading to inflammatory cell death.

Fig. R23 a Western blotting analysis was performed to evaluate the expression in 4T1 cells of GSDMD, GSDME, and MLKL upon treatment with different immune modulators and corresponding quantitative analyses of GSDMD (b), GSDME (c), MLKL (d) in 4T1 cells (n = 3 biological replicates). Statistical significance was evaluated using one-way ANOVA.

We sincerely appreciate you highlighting this limitation in our work. As you noted, since *Fn*-OMV is not a well-defined substance, it poses challenges for quality control in future applications. Based on your valuable feedback, in the revised manuscript we have discussed this limitation of the current study. Furthermore, in future work our aim is to explore identifying the active components responsible for the observed upregulation of PANoptosis component expression.

2. According to your valuable suggestion, we have also included additional immunoblot assays for ZBP1 and AIM2. The experimental results demonstrated that among these adjuvants, IFN-gamma and Poly(I:C) elevated the expression of both ZBP1 and AIM2, whereas PAM3CSK4 and *Fn*-OMV only increased the expression of ZBP1 with no significant effect on AIM2 expression, as illustrated in **Fig. R24**. Through the key point of PANoptosis executor

proteins, this study linked oHSV with *Fn*-OMV. On one hand, oHSV promotes the activation of PANoptosis executor proteins, while on the other, *Fn*-OMV upregulates the expression levels of PANoptosis executor proteins. The two possess distinct modes of action, yet they complement each other and generate positive feedback effects that amplify the therapeutic outcomes of oncolytic viruses.

Fig. R24 Cellular protein was analyzed by Western blotting to detect the expression level of ZBP1 and AIM2 in 4T1 cells treated with LPS- $0.5\mu\text{g ml}^{-1}$, IFN-gamma- 20ng ml^{-1} , Poly(I:C)- $1\mu\text{g ml}^{-1}$, IL-2- 10ng ml^{-1} , PAM3CSK4- $1\mu\text{g ml}^{-1}$, *Fn*-OMV- $1\mu\text{g ml}^{-1}$.

3. Based on your suggestion, we further tested the differences in synergistic ability against the antitumor effects of oHSV between LPS, Poly(I:C), IFN-gamma, PAM3CSK4 and *Fn*-OMV. The results showed that among LPS, Poly(I:C), IFN-gamma, and PAM3CSK4, *Fn*-OMV most significantly enhanced the anti-tumor effects of oHSV. The experimental results are depicted in the **Fig. R25**:

Fig. R25 a Schematic representation of 4T1 tumor inoculation and treatment with oHSV and different immune modulators (LPS- $5\mu\text{g}$, Poly(I:C)- $5\mu\text{g}$, IFN-gamma- $2\mu\text{g}$, PAM3CSK4- $5\mu\text{g}$, *Fn*-OMV- $5\mu\text{g}$) for the killing assay. **b** The tumor growth curve showed the influence of different immune modulators on the oHSV therapeutic effect ($n = 5$ biological replicates). Statistical significance was evaluated using two-way ANOVA.

Revision made:

In the revised manuscript, the following statements in Discussion were added:

*“The use of the *Fn*-OMV is a nice addition to the therapy. However, it is not a defined substance so quality control for patient use may prove challenging. In the future the active ingredients will likely require definition.”*

Minor Comment 1:

1. The authors should provide a specific definition for PANoptosis, as this is still a new term in the field and may not be well known to readers. PANoptosis is defined as a unique innate immune lytic, inflammatory programmed cell death pathway that is driven by caspases and

receptor interacting protein kinases (RIPKs) and regulated by multiprotein PANoptosome complexes (PMID: 36782083, 36253067). Additionally, the initial description of ZBP1 as an upstream sensor that activates PANoptosis occurred in PMID: 27917412, and this reference should be cited.

Response 1:

We sincerely appreciate your constructive suggestion. As you insightfully noted, "PANoptosis is a relatively new term that may not be familiar to all readers." Based on this valuable suggestion, we have included the definition of PANoptosis in the revised manuscript.

We are very grateful to you for pointing out our oversight. According to your suggestion, we have also cited the key references that initially described ZBP1 as an upstream sensor activating PANoptosis as recommended.

Revision made:

In the revised manuscript, the definition of PANoptosis was added in Introduction:

“PANoptosis is defined as a unique innate immune lytic, inflammatory programmed cell death pathway that is driven by caspases and receptor interacting protein kinases (RIPKs) and regulated by multiprotein PANoptosome complexes^{23, 24}.”

In the Introduction, we have cited the key references that initially described ZBP1 as an upstream sensor activating PANoptosis:

“Research has revealed that ZBP1 is a key upstream regulator of PANoptosis^{19, 29, 30}.”

Minor Comment 2:

2. The immunofluorescence for ZBP1 and Z-NA localization (Fig. 2) should be strengthened with additional controls. Nuclear localization of ZBP1 could be further confirmed by cellular fractionation to separate nuclear and cytosolic fractions followed by immunoblotting. Additionally, the Z-NA antibody specificity in recognizing Z-NAs and no other forms of nucleic acids should be confirmed. Details on the infection efficacy and viral titres in infected cells for Fig. 2f should also be provided.

Response 2:

We deeply appreciate your constructive recommendation. Following your advice, we have established negative and positive control groups and repeated the immunofluorescence validation for the localization of ZBP1 and Z-NA. For the positive control group, we treated cells with the small molecule drug CBL0137, which has been documented in multiple articles to promote the formation of Z-NA and upregulate ZBP1 expression (PMID: 35614224). The results are presented in **Fig. R26**.

Fig. R26 Z-NAs and ZBP1 expression in cells following oHSV infection and CBL0137 treatment. Immunofluorescence staining was employed to detect the expression of Z-NA and ZBP1 post-infection with oHSV (MOI = 1, 24 h). The negative control group consisted of untreated cells, while the positive control group was treated with CBL0137, a known inducer of Z-DNA.

To further ascertain the nuclear localization of ZBP1, we separated cytoplasmic and nuclear proteins for protein-level detection. Compared to the untreated group, the addition of oHSV significantly upregulated the expression of ZBP1 protein in the cell nucleus. The experimental results are depicted in **Fig. R27**.

In order to determine the specificity of the Z-NA antibody, we concurrently stained cells treated with oHSV with Z-NA and dsDNA antibodies for fluorescence imaging. The results revealed substantial differences in the nucleic acids recognized by the two, reinforcing the specificity of the Z-NA antibody for left-handed nucleic acids (shown in **Fig. R28**).

Fig. R28 Specificity of Z-NA antibody determined by immunofluorescent staining in oHSV-infected cells. Immunofluorescence staining for Z-NAs and dsDNA was performed to visualize their respective distributions post-infection with oHSV (MOI = 1, 24 h).

We have also supplemented the data on viral titers and infection efficiency. The virus titer used in the tests was 1×10^9 TU/mL (calculated via plaque assay). Post 12 hours of infection at various MOIs with GFP-labeled virus, we evaluated the infection efficiency of the cells (as shown in **Fig. R29**). Excessively high oncolytic virus treatment concentrations could harm healthy cells; model predictions indicate that low concentrations of oncolytic viral therapy can also achieve favorable antitumor effects (PMID: 23906157). In this study, we selected an MOI

of 1, which resulted in an infection efficiency of around 20%, and achieved a substantial cytotoxic effect 24 hours post-infection.

Fig. R29 Assessment of infection efficiency in cells exposed to GFP-oHSV. The infection efficiency of cells following a 12-hour exposure to GFP-labeled oHSV-1 at MOI of 0.1, 1, and 10.

We apologize for any inconvenience this might have caused you and our readers. We have carefully revised and supplemented the content, which we believe will present our research findings more effectively. Thank you again for your meticulous guidance.

Revision made:

In the revised manuscript, we have added **Fig. R27** as **Fig. 1**.

Fig. R27 h Expression levels of ZBP1 in cytoplasm and nucleus in 4T1 cells treated or untreated with oHSV were examined by Western blotting. **i** Western blotting was used to quantitatively analyze the expression of ZBP1 in cytoplasm and nucleus for 4T1 cells treated or untreated oHSV. n = 3 biological replicates. **j** Expression levels of ZBP1 in cytoplasm and nucleus in 4MOSC1 cells treated or untreated with oHSV were examined by Western blotting. **k** Western blotting was used to quantitatively analyze the expression of ZBP1 in cytoplasm and nucleus for 4T1 cells treated or untreated oHSV. n = 3 biological replicates.

In the revised manuscript, details on the viral tires in infected cells for Fig. 2f was added in figure legend:

“f The expression of Z-RNA in tumor cells after oHSV (MOI = 1, 24 h) treatment and IFNAR-1 blockade was evaluated using a Confocal Laser Scanning Microscope. Scale bars = 10 μm.”.

Minor Comment 3:

3. The authors show that Fn and Fn-OMV increase the expression of PANoptosis executioners

(Fig. 4); however, it is not clear whether the combination of oHSV with Fn or Fn-OMV results in a further increase in expression. This should be tested.

Response 3:

We sincerely appreciate your constructive comment. According to your valuable suggestion, we supplemented experiments examining the impact of combined oHSV and *Fn*-OMV treatment on the expression of PANoptosis executioners. The results showed that compared to oHSV alone, combination of *Fn*-OMV with oHSV led to higher expression levels of PANoptosis executioners (GSDMD-N, GSDME-N, p-MLKL), as depicted in **Fig. R30**.

Revision made:

The data in **Fig. R30** were added as **Supplementary Fig. 5** in the revised **Supplementary Information**.

Fig. R30 Cellular protein was analyzed by Western blotting to detect the expression level of PANoptosis related genes in 4T1 cells treated with *Fn*-OMV alone, oHSV alone or in combination and corresponding quantitative analyses of GSDMD-N, GSDME-N and p-MLKL (n = 3 biological replicates).

Minor Comment 4:

4. The authors should examine the effect of IFNAR1 neutralizing antibody treatment on cell death, ATP/LDH release, and NLRP6 protein expression in response to oHSV.

Response 4:

We sincerely appreciate your constructive suggestions, which has strengthened our study. According to your valuable suggestion, we supplemented the following experiments:

1. We examined the impact of IFNAR-1 neutralizing antibody treatment on cell death, which showed reduced oHSV-induced cell death upon IFNAR1 blockade, as depicted in **Fig. R31**.
2. We examined the impact of IFNAR-1 neutralizing antibody treatment on ATP/LDH release, which showed reduced oHSV-induced ATP/LDH release upon IFNAR-1 blockade, as depicted in **Fig. R32**.
3. We examined the impact of IFNAR-1 neutralizing antibody treatment on NLRP6

protein expression, which showed no significant effect on oHSV-induced upregulation of NLRP6, as depicted in **Fig. R33**.

Fig. R33 4T1 cells were infected with oHSV (MOI = 1) and IFNAR-1 blockade for 24 h, cellular protein was analyzed by Western blotting to detect the expression level and corresponding quantitative analyses of NLRP6 (n = 3 biological replicates).

Revision made:

In the revised manuscript, we have added **Fig. R31** as **Supplementary Fig. 2**.

Fig. R31 b The effect of IFNAR-1 on relative cell viability in the 4T1 cells treated with oHSV for 24 h (n = 3 biological replicates).

In the revised manuscript, we have added **Fig. R32** as **Fig. 2**.

Fig. R32 The release of LDH (left) and ATP (right) from 4T1 tumor cells after oHSV treatment and IFNAR-1 blockade (n = 3 biological replicates).

Minor Comment 5:

5. More methodological information is required for several of the analyses. In particular, the authors should explain how the protein expression profiles were obtained for each experiment (for example: Fig. 1a, c-e; Fig. 2b-d; Fig. 5g, i; and Fig. 6c, f), such as sequencing technique, viral dose, timepoints, etc. Additionally, the immunoblots in Fig. 4j-l should include a loading control.

Response 5:

We thank you for this important comment. We sincerely apologize that our description of experimental methods and details was lacking completeness, which likely caused confusion for you and other readers. Based on your valuable suggestion, we have provided more thorough methodological information for all experiments.

According to your valuable suggestion, we have also included a loading control for the immunoblots shown in Fig. 4j-l, as depicted in **Fig. R34**.

Thank you again for highlighting this important issue - it will ensure reproducibility and proper interpretation of our work. Please advise if any part still requires clarification.

Revision made:

The data of **Fig. R34** has been added in the manuscript as “**Fig. 4**”.

Fig. R34 Immunoprecipitation (IP) and Western blotting analysis were performed to examine the ubiquitination and expression of GSDMD (**j**), GSDME (**k**), and MLKL (**l**) in Control and Gsdmd, Gsdme, or Mlkl-depleted 4T1 cells after 24 hours of *Fn*-OMV treatment.

The more thorough methodological information was added in the Materials and Methods:

“For the detection of inflammasomes, select key innate sensor-related genes, and ISGs in tumor cells, we introduced oHSV with a viral titer of 1×10^9 TU/ml and an MOI of 1, and stimulated the cells for 24 hours. To assess the transition of M1/M2 macrophage polarization, Raw 264.7 cells were treated with oHSV (MOI = 0.1, 1, 5) or *Fn*-OMV at a concentration of 1 μ g/ml for 12 hours. For the evaluation of inflammatory gene expression in tumor tissues, the tissues were first homogenized using a cryo-grinding technique to obtain a consistent tissue lysate. Then tumor tissues or cultured cells were subjected to total RNA extraction following the manufacturer's instructions (Axygen). The concentration of RNA was determined using a spectrometer (Shimadzu UV-2401PC). Reverse transcription of one microgram of total RNA was carried out using HiScript II Reverse Transcriptase (Vazyme Biotech), following the manufacturer's protocol. Approximately 1% of the resulting cDNA was utilized as a template in each qRT-PCR reaction, performed with SYBR master mix (Vazyme Biotech). The qRT-PCR protocol employed three steps. The $2^{-\Delta\Delta CT}$ method was employed to calculate the fold induction, with each cDNA data point normalized to GAPDH expression. The primer sequences can be found in Supplementary Table 1. To effectively illustrate the qRT-PCR results, we utilized the heatmap function in R Studio software to plot the normalized gene expression data. The heatmap's color palette was strategically chosen to represent the continuum of expression levels: shades of blue denoted downregulated gene expression compared to the control, while a progression towards red signified upregulated expression. This method produced a visually intuitive gradient, facilitating an immediate perception of gene expression alterations.”

Minor Comment 6:

6. The authors should take care when discussing “expression” vs “activation” in the text. "Expression" does not always indicate "activation." For example, line 144, "blotting analysis confirmed the activation of ZBP1 and NLRP6...", should be amended.

Response 6:

We sincerely appreciate you pointing out our oversight. As you correctly noted, we need to precisely use and interpret the terminology of "expression" versus "activation". According to your suggestion, we have carefully re-examined the manuscript and made appropriate modifications where similar issues were identified.

Revision made:

In the revised manuscript, the term "protein activation" has been amended to "protein upregulation.":

“Western blotting analysis further confirmed the upregulation of ZBP1 and NLRP6 at the protein level in all five cell lines upon oHSV infection (Fig. 1c). Consistent with the findings from tumor cell lines, we also observed the upregulation of ZBP1 and NLRP6 in 4T1 tumors treated with oHSV (Fig. 1d).”.

“To further explore the potential role of elevated ZBP1 and NLRP6 expression in oHSV infection in tumor cells, we generated ZBP1-deficient 4T1 and 4MOSC1 cells.”.

“Our previous research revealed that oHSV induces cell death through the upregulation of ZBP1.”.

Reviewer #3

This paper claimed that combination of oncolytic viral (OV) therapy with bacterial outer membrane vesicles (OMVs) would potentiate the therapeutic efficacy of current OV-based modalities, which have shown limitations in clinical trials. The key claims made in this paper are 1) the authors revealed the detailed mechanism by which oncolytic HSV activates ZBP-1 mediated cancer cell death and 2) combination of oHSV with Fn-OMV can further increase antitumor efficacy. These claims are mostly supported by sufficient data. And most experiments were well designed and their analysis or interpretation on the data are fair. However, I cannot recommend this paper in its current state for publication and would consider a revised draft. There are some critical issues to be considered and addressed before publication, which are noted below.

Response:

We are truly appreciative of your constructive and positive comments. We also deeply appreciate your identification of areas in need of improvement and the constructive suggestions you have provided to enhance our research. In accordance with your advice, we have re-categorized and summarized the novel findings of our study. Furthermore, we have completed all the experiments you recommended and addressed all issues you have highlighted. All revisions have been distinctly marked in red in the revised manuscript and Supplementary Information for ease of identification. Once again, we thank you for your constructive feedback and for granting us the opportunity to revise our work.

Comment 1:

1. The findings that HSV can activate ZBP-1 and thus cause innate immune cytokine release, leading to infected cell death through inflammasome formation. In addition, the role of ZBP-1 pathway in cancer cell death is also known. In the context, it is natural to think that oHSV might induce cancer cell death through the ZBP-1-mediated pathway. What is a new finding or claim in this paper? Which can provide a new insight to the readers?

Response 1:

We sincerely appreciate your constructive comment. We apologize that our description was not comprehensive enough and did not sufficiently highlight the innovative aspects and practical value of this study, which likely caused confusion for you and other readers. Based on your suggestion, we have summarized the key innovations and potential applications of the current work:

1. This study provides the first evidence that oHSV elicits antitumor effects through the induction of ZBP1-mediated PANoptosis, and reveals for the first time that oHSV activates ZBP1 in tumor cells via the generation of endogenous Z-RNA:

Our results suggest that, in comparison to other nucleic acid sensors, ZBP1 may serve as a more sensitive detector involved in the therapeutic response to oHSV. It is known that HSV-1 activates the AIM2 inflammasome in BMDM, with ZBP1 and Pyrin acting downstream (PMID: 34471287). The current study reported for the first time that during HSV-1 infection, AIM2 expression was upregulated in some tumor cell lines (4MOSC1, 4MOSC2) but downregulated in others (SCC7, 4T1, CT26), while ZBP1 expression was upregulated across all five tumor cell lines tested. Further investigation revealed that ZBP1-mediated PANoptosis plays an important role in the antitumor effects of oHSV in

cancer. These findings characterize novel regulation of innate sensors during oHSV infection in tumor cells compared to previous reports in macrophages, and identify ZBP1-dependent PANoptosis as a key mechanism underlying the therapeutic activity of this oncolytic approach.

Previous studies have found that intracellular replication of certain viruses like influenza virus (IAV) and vaccinia virus (VACV) can generate substantial amounts of Z-RNA to activate ZBP1 (PMID: 32200799, 34192517). However, replication of other viruses such as rhabdovirus (VSV) or poliovirus (EMCV) does not produce detectable Z-RNA (PMID: 32200799). In this study, we have identified for the first time that oHSV primarily activates ZBP1 in tumor cells through the production of endogenous Z-RNA.

2. This study identified that *Fusobacterium nucleatum* outer membrane vesicles (*Fn*-OMV) can upregulate the expression of PANoptosis executor proteins by interfering with their ubiquitination:

OMVs can deliver various molecules (lipids, proteins, LPS and nucleic acids) to tumor cells, activating innate and adaptive immunity. This research identifies that *Fn*-OMV enhance the expression of PANoptosis executor proteins, such as GSDMD/E and MLKL, by interfering with their ubiquitination, showcasing a unique advantage of *Fn*-OMV over common immune adjuvants and providing a new insight into the adjunctive mechanism of *Fn*-OMV in oncolytic virus therapy. These executors (GSDMD/E, MLKL) play critical roles in tumor killing and immune activation, as their low expression can impair these effects. Low GSDME expression results in immune-silent apoptosis rather than immunogenic cell death upon stimulation (PMID: 28459430). High MLKL promotes antitumor activity and reduces metastasis (PMID: 30143632). Since oHSV exerts antitumor effects through ZBP1-mediated PANoptosis, *Fn*-OMV has excellent potential for combination with oHSV.

This study thus provides novel insight into the mechanism of *Fn*-OMV adjuvant activity and its synergistic use with oHSV.

3. This study introduces, for the first time, the strategy of combining *Fn*-OMV with oHSV and reveals its powerful antitumor immune effects:

Through the key point of PANoptosis executor proteins, this study linked oHSV with *Fn*-OMV. On one hand, oHSV promotes the activation of PANoptosis executor proteins, while on the other, *Fn*-OMV upregulates the expression levels of PANoptosis executor proteins. The two possess distinct modes of action, yet they complement each other and generate positive feedback effects that amplify the therapeutic outcomes of oncolytic viruses.

This study provides a highly feasible and effective clinical therapeutic approach for the future. The identification of *Fn*-OMV as a potent adjuvant sheds novel mechanistic insights and has significant implications for optimizing oncolytic virus.

Revision made:

In the revised manuscript, the new findings of this study have been incorporated into the Introduction and Discussion:

“This research suggests that oHSV may exert antitumor effects through the induction of ZBP1-mediated PANoptosis. Moreover, the study indicates that oHSV promotes the upregulation of ZBP1 expression in tumor cells by generating endogenous Z-RNA. The investigation also reveals that Fusobacterium nucleatum-OMV (Fn-OMV) can upregulate the expression of PANoptosis executor proteins (GSDMD/E, MLKL) by interfering with ubiquitination processes. Additionally, this study reveals that the combination strategy of Fn-OMV with oHSV harbors potent antitumor immune effects. Overall, through the key point of PANoptosis executor proteins, this study linked oHSV with Fn-OMV. On one hand, oHSV promotes the activation of PANoptosis executor proteins, while on the other, Fn-OMV upregulates the expression levels of PANoptosis executor proteins. The two possess distinct modes of action, yet they complement each other and generate positive feedback effects that amplify the therapeutic outcomes of OVs.”.

“OVs have emerged as a promising anticancer modality, albeit with recognized constraints as monotherapies¹. Addressing the need to augment their lytic efficacy is therefore of paramount importance. Our findings suggest that oHSV-induced interferon response within tumor cells facilitates ZBP1-driven PANoptosis by elevating Z-RNA levels. Based on these findings and our previous research results, we screened and identified Fn-OMVs that can enhance the expression levels of PANoptosis execution proteins GSDMD/E and MLKL. This combination strategy, encompassing Fn-OMVs and oHSV, bolstered inflammatory responses and exhibited profound antitumor immune activation in vivo, transforming immunologically "cold" tumors "hot" and heightening the effectiveness of ICB.”.

Comment 2:

2. A key claim in this paper is that ISG mRNAs undergo conformational changes and are converted to corresponding Z-RNAs. However, no solid evidence for this claim was provided. According to a previous literature (Ref #21), it said that Z-type dsRNA structures could be enriched in the 3'-UTR of ISG mRNAs. In this paper, ISG mRNAs were highly upregulated by oHSV treatment in cancer cells. It is also natural to think that the upregulated ISG mRNAs that contain Z-RNA structures in their 3'-UTRs can induce ZBP-1 activation, not through conversion of mRNAs into Z-RNAs.

Response 2:

We sincerely apologize for the errors made in the description of our results and the depiction of our mechanism figure, which have caused confusion for both you and our readers. As you correctly indicated, oHSV induces the production of a large number of ISGs, and the elevation of ISGs containing 3'-UTR Z-prone regions is a significant source of intracellular Z-RNA.

Based on our current results, we posit that the upregulation of ISGs with 3'-UTR Z-prone regions results in an increase in cellular Z-RNA levels, and that ZBP1 is extremely sensitive to the presence of Z-conformations in nucleic acids, thus detecting these specific Z-structures within the ISGs' 3'-UTRs, leading to its upregulation. Following your suggestion, we have amended the relevant descriptions and revised our figures (**Fig. R35**). We are very thankful for your meticulous attention to this matter and for bringing this issue to our attention.

Revision made:

In the revised manuscript, the “**Fig. 2g**” was replaced to the figure as shown in **Fig. R35**:

Fig. R35. oHSV upregulates ZBP1 by inducing the accumulation of Z-RNA. The schematic illustrates oHSV infection induces the upregulation of ISGs, whose 3' UTR regions form Z-RNA structures, thereby activating and increasing the expression of ZBP1.

In the revised manuscript, the description of experiments *in vitro* in “**Fig. 2**” was changed to:

“We examined the expression of ISG mRNAs in 4T1 and 4MOSC1 cells during oHSV infection and observed a significant upregulation of ISG mRNAs, along with an elevation of ISG mRNAs containing Z-prone regions within 3'-UTR (Fig. 2a, b).”

Comment 3:

3. In relation to the above comment 2, in Fig. 2a and 2f, the authors used anti-Z-DNA antibody and detected Z-DNA, NOT Z-RNA, in nucleus of the treated cancer cells. As noted above, the upregulation of ISG mRNAs itself can result in ZBP-1 activation. If so, why did the authors detect generation of Z-DNA in nucleus? What is direct correlation between the detection (presence) of Z-DNA in nucleus and ISG mRNAs? Then, what is a new finding in this paper?

Response 3:

We extend our sincere gratitude to you for pointing out our mistake. The antibody in question (Anti-Z-DNA, Ab00783-3.0, Absolute Antibody) was formerly known as Anti-Z-DNA (Clone Z22 mAb) [Z22]. However, multiple significant publications within the field have confirmed its efficacy in detecting Z-RNA (PMID: 34192517, 32200799). The manufacturer’s official website has also recently updated the product description, which now states the ability of the antibody to simultaneously detect both Z-DNA and Z-RNA. Accordingly, the antibody has been renamed to Anti-Z-DNA/Z-RNA (Clone Z22 mAb). In light of this, the antibody was used in our research to detect total left-handed nucleic acids (Z-DNA and Z-RNA). We deeply regret the oversight in not updating the name of the antibody and for the incorrect description in our manuscript, which may have caused confusion to you and the readers.

We examined the expression of ISG mRNAs in 4T1 and 4MOSC1 cells during oHSV infection and observed a significant upregulation of ISG mRNAs, along with an elevation of ISG mRNAs containing Z-prone regions within 3'-UTR (**Fig. R36a, b**). Furthermore, when the IFNAR-1 was blocked using neutralizing antibodies, the expression of ISG mRNAs no longer increased (**Fig. R36c**). Western blotting analysis confirmed that the addition of neutralizing antibodies completely inhibited the expression of ZBP1 (**Fig. R36d**). The predominant method currently employed for the detection of Z-nucleic acids is through fluorescence observation techniques. To further substantiate that the Z-nucleic acids emanating from ISGs are mainly Z-

RNA as opposed to Z-DNA, in accordance with your suggestion, we conducted supplementary experiments as follows:

We initiated our protocol by incorporating DNase I to decompose DNA, followed by fluorescent detection, which revealed that DNase I treatment did not substantially alter the fluorescent signals of the Z-nucleic acids. This indicates that the accumulation of Z-nucleic acids is largely attributed to Z-RNA rather than Z-DNA (Fig. R36e, f).

Fig. R36 oHSV activates ZBP1 by inducing the accumulation of Z-RNA. **a** Heatmap showing the expression of signature genes related to ISG mRNA in oHSV-treated (MOI = 1, 24 h) 4T1 (left) and 4MOSC1 (right) cells. $n = 3$ biological replicates. **b** Heatmap indicated the expression of signature genes related to ISG mRNA with high editing indices in oHSV-treated (MOI = 1, 24 h) 4T1 (left) and 4MOSC1 (right) cells. $n = 3$ biological replicates. **c** Heatmap demonstrated the influence of the expression of ISG mRNA with high editing indices on oHSV-treated (MOI = 1, 24 h) 4T1 cells after IFNAR-1 blockade. $n = 3$ biological replicates. **d** 4T1 cells were infected with oHSV (MOI = 1, 24 h) and IFNAR-1 blockade, cellular protein was analyzed by Western blotting to detect the expression level of ZBP1. **e** The expression of Z-NA and ZBP1 in 4T1 tumor cells after oHSV (MOI = 1, 24 h) alone and oHSV combined with DNase I treatment was evaluated using a Confocal Laser Scanning Microscope. Scale bars =

10 μm . **f** The expression of Z-RNA in tumor cells after oHSV (MOI = 1, 24 h) treatment and IFNAR-1 blockade was evaluated using a Confocal Laser Scanning Microscope. Scale bars = 10 μm .

We express our profound gratitude for your meticulous correction on this matter, which is instrumental in rigorously refining our study. Your feedback is greatly valued.

The three new findings from the study are:

1. ZBP1-Mediated PANoptosis Elicited by oHSV in Tumor Cells: The study reveals novel regulatory patterns of innate sensors in tumor cells during oHSV infection, with the first-time observation that oHSV variably modulates AIM2 expression across different tumor cell lines, while consistently upregulating ZBP1, which is pivotal for its antitumor efficacy via ZBP1-mediated PANoptosis. The study is the first to reveal that oHSV primarily activates ZBP1 in tumor cells through the production of endogenous Z-RNA.
2. F_n-OMV in Upregulating PANoptosis Executor Proteins: This research identifies that *F_n*-OMV enhance the expression of PANoptosis executor proteins, such as GSDMD/E and MLKL, by interfering with their ubiquitination, showcasing a unique advantage of *F_n*-OMV over common immune adjuvants and providing a new insight into the adjunctive mechanism of *F_n*-OMV in oncolytic virus therapy.
3. Novel Combination Strategy for Enhanced Antitumor Immunity: For the first time, the study presents the strategy of combining *F_n*-OMV with oHSV, demonstrating powerful antitumor effects. The combination results in differential yet complementary activation and expression upregulation of PANoptosis executors, culminating in a synergistic enhancement of the therapeutic efficacy of oncolytic viruses, highlighting the potential for translation into a feasible and effective clinical therapeutic approach.

Revision made:

In the revised manuscript, Fig. R37 has been added as Fig. 2:

Fig. R37 The expression of Z-NA and ZBP1 in 4T1 tumor cells after oHSV (MOI = 1, 24 h) alone and oHSV combined with DNase I treatment was evaluated using a Confocal Laser Scanning Microscope. Scale bars = 10 μm .

In the revised manuscript, the following statements were added in the Results section:

“We examined the expression of ISG mRNAs in 4T1 and 4MOSC1 cells during oHSV infection and observed a significant upregulation of ISG mRNAs, along with an elevation of

ISG mRNAs containing Z-prone regions within 3'-UTR (Fig. 2a, b). Furthermore, when the IFNAR-1 was blocked using neutralizing antibodies, the expression of ISG mRNAs no longer increased (Fig. 2c). Western blotting analysis confirmed that the addition of neutralizing antibodies completely inhibited the expression of ZBP1 (Fig. 2d). We initiated our protocol by incorporating DNase I to decompose DNA, followed by fluorescent detection, which revealed that DNase I treatment did not substantially alter the fluorescent signals of the Z-nucleic acids. This indicates that the accumulation of Z-nucleic acids is largely attributed to Z-RNA rather than Z-DNA (Fig. 2e).”.

Comment 4:

4. Although *Fn*-OMVs showed meaningful synergistic effects with oHSV, the preparation of OMVs in general is highly costly and has limitation in mass production. In contrast, heat-treated dead bacteria itself is readily and easily prepared in large scale. In the context, the authors need to compare the therapeutic efficacy of oHSV + *Fn*-OMV with that of oHSV + heat-treated dead *Fn*. I would expect there is not much difference in the efficacy.

Response 4:

Thank you for your valuable comment addressing the practical aspects of our therapeutic strategy. We concur that the production of OMVs, including *Fn*-OMVs, presents financial and scalability challenges, which contrasts with the relative ease of generating heat-killed bacteria.

In accordance with your suggestion, we have supplemented our study with experiments comparing the therapeutic effects of oHSV+*Fn*-OMV treatment to that of oHSV + heat-treated dead *Fn*. As you anticipated, there was no significant difference in efficacy (**Fig. R38**).

Fig. R38 **a** Schematic representation of 4T1 tumor inoculation and treatment with oHSV combined with heat-treated dead *Fn* (5×10^6 CFU) or *Fn*-OMV ($5 \mu\text{g}$) for the killing assay. **b** Average 4T1 tumor growth curve depicted as mean \pm s.e.m. for each treatment group ($n = 5$ biological replicates). Statistical significance was evaluated using two-way ANOVA.

We thank you for bringing this important issue to our attention. Based on your advice, we have included a discussion of this topic in the revised manuscript. heat-treated dead bacteria can serve as vaccines, be used for immunological research, or act as experimental models, triggering host immune responses due to their retained antigens. OMVs, containing a diverse array of bacterial components, are frequently used as adjuvants in vaccine design or to study bacterial-host cell interactions, and are also being explored as drug delivery systems. OMVs are often considered safer since they do not contain a complete bacterial genome.

Both heat-treated dead bacteria and OMVs have their own advantages and research and application values. Given the current controversy surrounding the roles of *Fn* in tumorigenesis,

where *Fn* is considered a contributing factor to certain tumors, we have cautiously chosen to investigate *Fn*-OMV from a safety perspective. Comparing the advantages and disadvantages of heat-killed *Fn* and *Fn*-OMV requires extensive future work. We are grateful once again for the significant issue you highlighted and, following your suggestion, have discussed it in the revised manuscript.

Revision made:

In the revised manuscript, the following statements were added in the Discussion section:

*“Both heat-killed bacteria and OMVs possess distinct advantages and research applications. OMVs production is generally costly and faces scalability challenges in mass production. In contrast, heat-killed bacteria are readily amenable to large-scale preparation. Heat-killed bacteria can serve as vaccines for immunological research or as experimental models to elicit host immune responses due to their preserved antigens. OMVs, containing a variety of bacterial components, are often used in vaccine design or adjuvants for studying bacterial-host cell interactions, as well as in drug delivery systems. OMVs are typically considered safer as they do not encompass complete bacterial genomes⁵¹. Given the current controversy surrounding *Fn* role in tumorigenesis and its association with certain tumors as a potential risk factor, we cautiously chose to investigate *Fn*-OMVs from a safety perspective. Comparative analysis of the pros and cons of heat-killed *Fn* and *Fn*-OMVs necessitates extensive future work.”*

Comment 5:

5. Z-NA seems not accurate expression. Instead, Z-DNA or Z-RNA depending on situation should be used at right place.

Response 5:

Thank you for your attention to detail in our manuscript. The use of "Z-NA" is not specific and may lead to ambiguity. Throughout the manuscript, we will ensure to use the precise term "Z-RNA" depending on certain situation. The term "Z-NA" is employed to describe instances where nucleic acid structures may include both Z-DNA and Z-RNA.

Revision made:

In the revised manuscript, the “Fig. 2f” was replaced to the figure as shown in Fig. R39

Fig. R39 Immunofluorescence microscopy was performed to detect the expression of Z-RNA in different treatment groups

In the revised manuscript, the following statements were corrected:

“Mechanistically, oHSV enhances the expression of interferon-stimulated genes (ISGs),

leading to the accumulation of endogenous Z-RNA and subsequent activation of ZBP1. oHSV activates ZBP1 by inducing the accumulation of Z-RNA. Our findings suggest that oHSV-induced interferon response within tumor cells facilitates ZBP1-driven PANoptosis by elevating Z-RNA levels.”

Comment 6:

6. There are many abbreviations without spelling out or definition at their first appearance; for examples, PANoptosis, OSCC, etc. A table or note for these abbreviations could be made separately.

Response 6:

Thank you for your comment on the abbreviations used in our manuscript. We apologize for any confusion caused by the lack of spelled-out terms or definitions for abbreviations like PANoptosis, OSCC, and others upon their first occurrence. To address this, we have now included a note listing all abbreviations along with their full terms and definitions. This note is positioned at the end of the manuscript. We have also ensured that each abbreviation is spelled out and defined in the text at its first use.

Revision made:

We have added the abbreviation note in the manuscript:

“PANoptosis: pyroptosis, apoptosis, necroptosis; OV: oncolytic viruses; HSV-1: herpes simplex virus-1; oHSV: HSV-1-based OVs (oHSV); OMV: outer membrane vesicles; Fn: *Fusobacterium nucleatum*; ICB: immune checkpoint blockade; RIPKs: receptor interacting protein kinases; ZBP1: Z-DNA binding protein 1; Z-NA: Z-type nucleic acids; MLKL: mixed lineage kinase domain-like protein; GSDMD/E: gasdermin D/E; AIM2: absent in melanoma 2; ISG: interferon-stimulated gene; qRT-PCR: quantitative real-time polymerase chain reaction; ADAR1: adenosine deaminase RNA specific 1; EREs: endogenous retroviral elements; OSCC: oral squamous cell carcinoma; TDLNs: tumor draining lymph nodes; ELISA: enzyme-linked immunosorbent assay; TEM: transmission electron microscope; DLS: dynamic light scattering; IHC: immunohistochemistry; Tregs: regulatory T cells; TILs: tumor-infiltrating lymphocytes; TNF- α : tumor necrosis factor alpha; HMGB1: high mobility group box 1.”

REVIEWERS' COMMENTS

Reviewer #1 (Remarks to the Author):

Wang et al. have provided an in depth and well-reasoned response to review comments with a wealth of new experimental details and additional useful experimental results. The novelty issue raised by R3 was adequately addressed by pointing out that this study provides convincing evidence that the anti-tumor activity of oHSV depends in part on the induction of ZBP1 that in turn induces PANoptosis and that ZBP1 activation is the result of Z-RNA generation. The Z-RNA levels appear to be derived from ISG containing 3'-UTR Z-prone regions. Thus, ZBP1 induction contributes to HSV-mediated tumor killing. A second important finding was that Fn-OMV vesicles can add to anti-tumor activity by reducing the ubiquitination of pyroptosis/necroptosis execution proteins such as GSDMD/E and MLKL. Admittedly this could be difficult for use in patients since Fusobacteria would be the source of the vesicles and it is difficult to define and purify the active molecules. If possible, it would be very interesting if the vector could be armed with the secret sauce. In response to R1, the authors have performed additional experiments to confirm the development of CD8 dependent anti-tumor immunity and abscopal activity. Overall the figures and legends are improved and easier to understand.

Reviewer #2 (Remarks to the Author):

The authors have sufficiently addressed my comments and adequately improved the manuscript. However, a few additional editorial updates would be helpful for clarity:

1. The authors currently define PANoptosis as an abbreviation for pyroptosis, apoptosis, and necroptosis, but this is inaccurate. PANoptosis is a distinct pathway, rather than a combination of other pathways.
2. The authors cite PMID: 34471287 to say that "HSV-1 activates the absent in melanoma 2 (AIM2) inflammasome in bone marrow-derived macrophages (BMDMs), with ZBP1 and Pyrin acting downstream." However, this is misleading, as HSV-1 activates the AIM2 PANoptosome with ZBP1 and Pyrin, rather than the AIM2 inflammasome.

Reviewer #3 (Remarks to the Author):

Most of critical issues and concerns were well addressed in this revision, and thus no further comments remain. I would recommend publication of this revised manuscript as is.

Point-to-point responses to the comments

Reviewer #1

Wang et al. have provided an in depth and well-reasoned response to review comments with a wealth of new experimental details and additional useful experimental results. The novelty issue raised by R3 was adequately addressed by pointing out that this study provides convincing evidence that the anti-tumor activity of oHSV depends in part on the induction of ZBP1 that in turn induces PANoptosis and that ZBP1 activation is the result of Z-RNA generation. The Z-RNA levels appear to be derived from ISG containing 3'-UTR Z-prone regions. Thus, ZBP1 induction contributes to HSV-mediated tumor killing. A second important finding was that Fn-OMV vesicles can add to anti-tumor activity by reducing the ubiquitination of pyroptosis/necroptosis execution proteins such as GSDMD/E and MLKL. Admittedly this could be difficult for use in patients since Fusobacteria would be the source of the vesicles and it is difficult to define and purify the active molecules. If possible, it would be very interesting if the vector could be armed with the secret sauce. In response to R1, the authors have performed additional experiments to confirm the development of CD8 dependent anti-tumor immunity and abscopal activity. Overall the figures and legends are improved and easier to understand.

Response:

We are immensely grateful for your recognition of our work and deeply appreciate the valuable time you have dedicated to reviewing our work and offering a wealth of constructive suggestions.

Reviewer #2

The authors have sufficiently addressed my comments and adequately improved the manuscript. However, a few additional editorial updates would be helpful for clarity

Response:

We are deeply grateful for your recognition of our work. Regarding the editorial updates you suggested, we have made adjustments in accordance with your recommendations. Once again, we thank you profoundly for dedicating your valuable time to review our work and for providing numerous invaluable suggestions.

Comment 1:

1. The authors currently define PANoptosis as an abbreviation for pyroptosis, apoptosis, and necroptosis, but this is inaccurate. PANoptosis is a distinct pathway, rather than a combination of other pathways.

Response 1:

We sincerely appreciate your constructive comments. As you pointed out, our initial depiction of PANoptosis as merely an abbreviation amalgamating pyroptosis, apoptosis, and necroptosis was inaccurate. According to your valuable suggestion, we have redefined PANoptosis accordingly.

Revision made:

In the revised manuscript, the definition of PANoptosis was added in Introduction:

“PANoptosis is defined as a unique inflammatory programmed cell death pathway that is driven by caspases and receptor interacting protein kinases (RIPKs) and regulated by multiprotein PANoptosome complexes.”

Comment 2:

2. The authors cite PMID: 34471287 to say that “HSV-1 activates the absent in melanoma 2 (AIM2) inflammasome in bone marrow-derived macrophages (BMDMs), with ZBP1 and Pyrin acting downstream.” However, this is misleading, as HSV-1 activates the AIM2 PANoptosome with ZBP1 and Pyrin, rather than the AIM2 inflammasome.

Response 2:

We are very grateful to you for pointing out our oversight. According to your suggestion, we have revised the description of that sentence.

Revision made:

In the revised manuscript, the description of HSV-1 activates the AIM2 PANoptosome with ZBP1 and Pyrin was added in Discussion:

“It is known that HSV-1 activates the absent in melanoma 2 (AIM2) PANoptosome with ZBP1 and Pyrin in bone marrow-derived macrophages (BMDMs)¹⁹.”

Reviewer #3

Most of critical issues and concerns were well addressed in this revision, and thus no further comments remain. I would recommend publication of this revised manuscript as is.

Response:

We are immensely grateful for your recognition of our work and deeply appreciate the valuable time you have dedicated to reviewing our work and offering a wealth of constructive suggestions.